# Provably and Practically Efficient Adversarial Imitation Learning with General Function Approximation

**Tian Xu**[*,1,2,3], **Zhilong Zhang**[*,1,2,3], **Ruishuo Chen**[†,1,2], **Yihao Sun**[1,2], **and Yang Yu**[‡,1,2,3]

[1]National Key Laboratory for Novel Software Technology, Nanjing University, China
[2]School of Artificial Intelligence, Nanjing University, China
[3]Polixir.ai

## Abstract

As a prominent category of imitation learning methods, adversarial imitation learning (AIL) has garnered significant practical success powered by neural network approximation. However, existing theoretical studies on AIL are primarily limited to simplified scenarios such as tabular and linear function approximation and involve complex algorithmic designs that hinder practical implementation, highlighting a gap between theory and practice. In this paper, we explore the theoretical underpinnings of online AIL with general function approximation. We introduce a new method called optimization-based AIL (OPT-AIL), which centers on performing online optimization for reward functions and optimism-regularized Bellman error minimization for Q-value functions. Theoretically, we prove that OPT-AIL achieves polynomial expert sample complexity and interaction complexity for learning near-expert policies. To our best knowledge, OPT-AIL is the first provably efficient AIL method with general function approximation. Practically, OPT-AIL only requires the approximate optimization of two objectives, thereby facilitating practical implementation. Empirical studies demonstrate that OPT-AIL outperforms previous state-of-the-art deep AIL methods in several challenging tasks. [1]

## 1 Introduction

Sequential decision-making tasks are prevalent in real-world applications, where agents seek policies that maximize long-term returns. Reinforcement learning (RL) [49] provides a well-known framework for developing effective policies through trial and error. However, RL often necessitates carefully designed reward functions and typically requires millions of interactions with the environment to achieve satisfactory performance [36, 20]. In contrast, imitation learning (IL) offers a more sample-efficient approach to learning effective policies by mimicking expert demonstrations, bypassing the need for explicit reward functions. As a result, IL has gained popularity and demonstrated success in a wide range of real-world applications such as recommendation systems [11, 46] and generalist robot learning [10, 35].

IL encompasses two main categories of methods: behavioral cloning (BC) and adversarial imitation learning (AIL). BC employs supervised learning to directly infer expert policies from demonstration

---

∗: Equal contribution. Emails: `xut@lamda.nju.edu.cn` and `zhangzl@lamda.nju.edu.cn`.
†: Joined this work as an undergraduate student of School of Mathematics, Nanjing University.
‡: Corresponding author. Email: `yuy@nju.edu.cn`.
[1]The code is available at `https://github.com/LAMDA-RL/OPT-AIL`.

38th Conference on Neural Information Processing Systems (NeurIPS 2024).

data [39, 44, 9]. In contrast, AIL utilizes an adversarial learning process to replicate the expert's state-action distribution. This process involves the learner recovering an adversarial reward to maximize the policy value gap and subsequently learning a policy that minimizes this gap under the recovered reward. Building on these foundational principles, numerous practical algorithms have been developed [55, 27, 9, 22, 26, 16, 15, 34, 30], achieving significant empirical advancements.

From these empirical advances, a notable observation is that AIL often significantly outperforms BC [16, 26, 27, 15]. To better understand this phenomenon, recent research has focused on the theoretical underpinnings of AIL [61, 45, 63, 33, 64, 57], particularly in the online setting. This research examines both *expert sample complexity* (the number of expert trajectories required) and *interaction complexity* (the number of trajectories needed when interacting with the environment), both of which are crucial for practical applications. In the tabular setting, the best-known complexity result is achieved in [64]. They developed the MB-TAIL algorithm, which leverages advanced distribution estimation, achieving the expert sample complexity $\widetilde{\mathcal{O}}(H^{3/2}|\mathcal{S}|/\varepsilon)$ and interaction complexity $\widetilde{\mathcal{O}}(H^3|\mathcal{S}|^2|\mathcal{A}|/\varepsilon^2)$, where $|\mathcal{S}|$ and $|\mathcal{A}|$ are the state space size and action space size, respectively, $H$ is the horizon length and $\varepsilon$ is the desired value gap. Furthermore, [33, 57] investigated the AIL theory in the linear function approximation setting. Notably, the BRIG approach proposed in [57] uses linear regression for policy evaluation and achieves the expert sample complexity $\widetilde{\mathcal{O}}(H^2d/\varepsilon^2)$ and interaction complexity $\widetilde{\mathcal{O}}(H^4d^3/\varepsilon^2)$, where $d$ is the feature dimension. For a complete summary of related results, please refer to Table 1.

Despite substantial theoretical advances, there still exists a gap between theory and practice in AIL. First, prior theoretical analysis primarily focuses on restricted settings such as tabular [41, 45, 64] or linear function approximation [33, 57], which deviate from practice where AIL approaches often operate with general function approximation (e.g., neural network approximation). Besides, most previous theoretical works involve algorithmic designs such as count-based [45, 64] or covariance-matrix-based [33, 57] bonuses, which are tailored to their respective settings. Implementing such algorithmic designs in practical settings, where neural network approximation is employed, presents significant challenges [65, 54].

Table 1: A summary of the expert sample complexity and interaction complexity. Here $H$ is the horizon length, $\varepsilon$ is the desired imitation gap, $|\mathcal{S}|$ is the state space size, $|\mathcal{A}|$ is the action space size, $|\Pi|$ is the cardinality of the finite policy class $\Pi$, $d$ is the dimension of the feature space, $d_{\mathrm{GEC}}$ is the generalized eluder coefficient, $\mathcal{N}(\mathcal{R}_h)$ and $\mathcal{N}(\mathcal{Q}_h)$ are the covering numbers of the reward class $\mathcal{R}_h$ and Q-value class $\mathcal{Q}_h$, respectively. We use $\widetilde{\mathcal{O}}$ to hide logarithmic factors.[2]

| Setting | Algorithm | Expert Sample Complexity | Interaction Complexity |
|---|---|---|---|
| General Function Approximation | BC [13] [3] | $\widetilde{\mathcal{O}}\left(\frac{H^3 \log(\max_{h\in[H]} |\Pi_h|)}{\varepsilon^2}\right)$ | 0 |
| Tabular MDPs | OAL [45] | $\widetilde{\mathcal{O}}\left(\frac{H^2|\mathcal{S}|}{\varepsilon^2}\right)$ | $\widetilde{\mathcal{O}}\left(\frac{H^4|\mathcal{S}|^2|\mathcal{A}|}{\varepsilon^2}\right)$ |
| Tabular MDPs | MB-TAIL [64] | $\widetilde{\mathcal{O}}\left(\frac{H^{3/2}|\mathcal{S}|}{\varepsilon}\right)$ | $\widetilde{\mathcal{O}}\left(\frac{H^3|\mathcal{S}|^2|\mathcal{A}|}{\varepsilon^2}\right)$ |
| Linear Mixture MDPs | OGAIL [33] | $\widetilde{\mathcal{O}}\left(\frac{H^3d^2}{\varepsilon^2}\right)$ | $\widetilde{\mathcal{O}}\left(\frac{H^4d^3}{\varepsilon^2}\right)$ |
| Linear MDPs | BRIG [57] | $\widetilde{\mathcal{O}}\left(\frac{H^2d}{\varepsilon^2}\right)$ | $\widetilde{\mathcal{O}}\left(\frac{H^4d^3}{\varepsilon^2}\right)$ |
| General Function Approximation | OPT-AIL | $\widetilde{\mathcal{O}}\left(\frac{H^2 \log(\max_{h\in[H]} \mathcal{N}(\mathcal{R}_h))}{\varepsilon^2}\right)$ | $\widetilde{\mathcal{O}}\left(\frac{H^4 d_{\mathrm{GEC}} \log(\max_{h\in[H]} \mathcal{N}(\mathcal{Q}_h)\mathcal{N}(\mathcal{R}_h))+H^2}{\varepsilon^2}\right)$ |

**Contribution.** This paper aims to bridge the gap between theory and practice in AIL by developing a provably efficient algorithm with general function approximation and providing a practical implementation equipped with neural networks.

First, we introduce a new AIL approach called optimization-based adversarial imitation learning (OPT-AIL) and provide a comprehensive theoretical analysis for general function approximation. The core of OPT-AIL involves minimizing two key objectives. To recover the reward, OPT-AIL solves an online optimization problem using a no-regret approach. For policy learning, inspired by [32], OPT-AIL infers the Q-value functions by minimizing the optimism-regularized Bellman

---

[2]We will not omit $\log(\mathcal{N}(\mathcal{F}))$ for a function class $\mathcal{F}$ in the $\widetilde{\mathcal{O}}$ notation since $\log(\mathcal{N}(\mathcal{F}))$ could not be small for many function classes.

[3]Here we present the result of BC in the worst case, which is consistent with this paper.

error and then derives the corresponding greedy policies. Under mild assumptions, we prove that OPT-AIL achieves the expert sample complexity $\widetilde{\mathcal{O}}(H^2 \log(\max_{h \in [H]} \mathcal{N}(\mathcal{R}_h))/\varepsilon^2)$ and interaction complexity $\widetilde{\mathcal{O}}((H^4 d_{\mathrm{GEC}} \log(\max_{h \in [H]} \mathcal{N}(\mathcal{Q}_h)\mathcal{N}(\mathcal{R}_h)) + H^2)/\varepsilon^2)$. Here $d_{\mathrm{GEC}}$ is the generalized eluder coefficient, originally proposed in [68] to measure the complexity of RL with function approximation, which we adapt to the AIL setting. $\mathcal{N}(\mathcal{R}_h)$ and $\mathcal{N}(\mathcal{Q}_h)$ are the covering numbers of the reward class $\mathcal{R}_h$ and Q-value class $\mathcal{Q}_h$, respectively. To our best knowledge, OPT-AIL is the first provably efficient AIL approach with general function approximation.

Furthermore, we offer a practical implementation of OPT-AIL, demonstrating its competitive performance on standard benchmarks. Notably, OPT-AIL only requires the approximate optimization of two objectives, simplifying its practical implementation with deep neural networks. Leveraging this advantage, we implement OPT-AIL using neural network approximations and compare its performance against prior state-of-the-art (SOTA) deep AIL methods, which often lack theoretical guarantees. Experimental results indicate that OPT-AIL outperforms SOTA deep AIL approaches on several challenging tasks within the DMControl benchmark.

## 2 Related Works

**Adversarial Imitation Learning.** The theoretical foundations of AIL have been extensively explored in numerous studies [1, 52, 48, 60, 67, 41, 45, 33, 40, 63, 50, 56, 64, 57]. Early research [1, 52, 48, 41, 60, 67, 62, 50, 56] focused on ideal settings where the transition function is known or an exploratory data distribution is available, primarily addressing expert sample efficiency. Notably, under mild conditions, [63] proved that AIL can achieve a horizon-free imitation gap bound $\mathcal{O}(\min\{1, \sqrt{|\mathcal{S}|/N}\})$, where $N$ denotes the number of expert trajectories. Recently, a new research direction has emerged that addresses more practical scenarios, specifically online AIL with unknown transitions [45, 33, 64, 57]. This line of work investigates both expert sample complexity and interaction complexity. These recent advancements were discussed in the previous section and thus will not be reiterated here. Most existing theoretical works focus on either tabular [41, 45, 64] or linear function approximation settings [33, 57], and often lack practical implementations due to algorithmic designs tailored to specific settings. In contrast, this work simultaneously provides theoretical guarantees for general function approximation and offers a practical implementation that demonstrates competitive performance.

On the empirical side, there has been extensive research [19, 27, 28, 16, 26, 15] developing practical AIL approaches that leverage general function (or neural network) approximation. A seminal method in this field is generative adversarial imitation learning (GAIL) [19]. In GAIL, a discriminator is trained to distinguish between samples from expert demonstrations and those generated by a policy, while the policy (or generator) learns to maximize the reward signal provided by the discriminator. More recently, [15] proposed inverse Q-Learning (IQLearn), which achieves SOTA performance across a diverse set of tasks. However, these practical methods often lack rigorous theoretical guarantees for general function approximation.

**General Function Approximation in Reinforcement Learning.** Our work is closely related to a body of research focused on general function approximation in RL [38, 21, 59, 23, 32]. Notably, [32] proposed an algorithmic framework that incorporates a unified objective to balance exploration and exploitation in RL, demonstrating a sublinear regret bound. In this paper, we adapt this algorithmic design to address several RL sub-problems within the context of AIL. Unlike the RL setting, where a fixed reward is provided in advance, AIL involves inferring the reward function from expert demonstrations and environment interactions collected by the learning policies. Therefore, our work requires developing a theoretical analysis for the joint learning process of both rewards and policies, highlighting a unique challenge in AIL compared to traditional RL.

## 3 Preliminaries

**Markov Decision Process.** In this paper, we consider episodic Markov Decision Processes (MDPs), represented by the tuple $\mathcal{M} = (\mathcal{S}, \mathcal{A}, P, r^{\mathrm{true}}, H, s_1)$. Here, $\mathcal{S}$ and $\mathcal{A}$ denote the state and action spaces, respectively. $H$ signifies the planning horizon, while $s_1$ stands for the fixed initial state. The set $P = \{P_1, \ldots, P_H\}$ characterizes the non-stationary transition function of this MDP. Specifically, $P_h(s_{h+1}|s_h, a_h)$ determines the probability of transiting to state $s_{h+1}$ given state $s_h$ and action $a_h$ at time step $h$, where $h \in [H]$. Similarly, $r^{\mathrm{true}} = \{r_1^{\mathrm{true}}, \ldots, r_H^{\mathrm{true}}\}$ outlines the reward function of this MDP. Without loss of generality, we assume $r_h^{\mathrm{true}} : \mathcal{S} \times \mathcal{A} \to [0, 1]$ for $h \in [H]$. A non-stationary

policy is denoted by $\pi = \{\pi_1, \ldots, \pi_H\}$ with $\pi_h : \mathcal{S} \to \Delta(\mathcal{A})$, where $\Delta(\mathcal{A})$ denotes the probability simplex. Here, $\pi_h(a|s)$ represents the probability of selecting action $a$ in state $s$ at time step $h$, for $h \in [H]$.

The quality of policy $\pi$ is evaluated by policy value:

$$V^\pi = \mathbb{E}\left[\sum_{h=1}^{H} r_h^{\text{true}}(s_h, a_h) \middle| a_h \sim \pi_h(\cdot|s_h), s_{h+1} \sim P_h(\cdot|s_h, a_h), \forall h \in [H]\right].$$

We denote the Q-value function of policy $\pi$ at time step $h$ as $Q_h^\pi : \mathcal{S} \times \mathcal{A} \to \mathbb{R}$, where $Q_h^\pi(s, a) = \mathbb{E}_\pi[\sum_{\ell=h}^{H} r_\ell^{\text{true}}(s_\ell, a_\ell)|s_h = s, a_h = a]$. The optimal Q-value function $Q_h^\star : \mathcal{S} \times \mathcal{A} \to \mathbb{R}$ is defined as $Q_h^\star(s, a) := \sup_{\pi \in \Pi} Q_h^\pi(s, a)$. It is known that $Q_h^\star$ is the fixed point of Bellman operator $\mathcal{T}_h$: $Q_h^\star(s, a) = (\mathcal{T}_h Q_{h+1}^\star)(s, a) := r_h^{\text{true}}(s, a) + \mathbb{E}_{s' \sim P_h(\cdot|s,a)}[\max_{a' \in \mathcal{A}} Q_{h+1}^\star(s', a')]$. In other words, $Q^\star$ has zero Bellman error, i.e., $Q_h^\star(s, a) - (\mathcal{T}_h Q_{h+1}^\star)(s, a) = 0$.

**Imitation Learning.** The essence of IL lies in acquiring a high-quality policy *without* the reward function $r^{\text{true}}$. In pursuit of this objective, we typically posit the existence of a near-optimal expert policy $\pi^{\text{E}}$ capable of interacting with the environment to generate a dataset, comprising $N$ trajectories each of length $H$: $\mathcal{D}^{\text{E}} = \{\tau = (s_1, a_1, s_2, a_2, \ldots, s_H, a_H)\,; a_h \sim \pi_h^{\text{E}}(\cdot|s_h), s_{h+1} \sim P_h(\cdot|s_h, a_h), \forall h \in [H]\}$. Subsequently, the learner leverages this dataset $\mathcal{D}^{\text{E}}$ to mimic the behavior of the expert and thereby derives an effective policy. The quality of this imitation is measured by the *imitation gap* [1, 43, 41]: $V^{\pi^{\text{E}}} - V^\pi$, where $\pi$ represents the learned policy. Essentially, we hope that the learned policy can perfectly mimic the expert such that the imitation gap is small.

AIL is a prominent class of IL methods that imitates expert behavior through an adversarial learning process defined by $\min_\pi \max_r V_r^{\pi^{\text{E}}} - V_r^\pi$, where $V_r^\pi$ denotes the value of policy $\pi$ under reward $r$. In this framework, AIL infers a reward function that maximizes the value gap between the expert policy and the learning policy. Subsequently, it learns a policy that minimizes this value gap using the inferred reward. Essentially, AIL involves solving several RL sub-problems, as the outer optimization problem concerning the policy is equivalent to an RL problem under the inferred reward $r$.

**General Function Approximation.** This work considers AIL with general function approximation. In this setup, the learner first has access to a reward class $\mathcal{R} = \mathcal{R}_1 \times \mathcal{R}_2 \times \ldots \times \mathcal{R}_H$ with $\forall h \in [H], \mathcal{R}_h \subseteq (\mathcal{S} \times \mathcal{A} \to [0, 1])$ to infer the reward. We assume that $\mathcal{R}$ captures the unknown true reward.

**Assumption 1** (Realizability of $\mathcal{R}$). *The unknown true reward lies in the reward class, i.e., $r^{\text{true}} \in \mathcal{R}$.*

Besides, the learner also has access to a Q-value function class $\mathcal{Q} = \mathcal{Q}_1 \times \mathcal{Q}_2 \times \ldots \times \mathcal{Q}_H$ with $\forall h \in [H], \mathcal{Q}_h \subseteq (\mathcal{S} \times \mathcal{A} \to [0, H])$, which is used for solving several RL sub-problems under different inferred rewards in AIL. Since there is no reward in the $H+1$ step, we always set $Q_{H+1} \equiv 0$. Below, we present two standard assumptions about the function class $\mathcal{Q}$ that are commonly adopted in the literature of RL with function approximation [59, 23, 68, 32].

**Assumption 2** (Realizability of $\mathcal{Q}$). *For reward $r \in \mathcal{R}$, $Q^{\star,r} \in \mathcal{Q}$, where $Q^{\star,r}$ denotes the optimal Q-value function under reward $r$.*

**Assumption 3** (Bellman Completeness of $\mathcal{Q}$). *For reward $r \in \mathcal{R}$, $\mathcal{T}_h^r \mathcal{Q}_{h+1} \subseteq \mathcal{Q}_h$, $\forall h \in [H]$, where $\mathcal{T}_h^r$ denotes the Bellman operator under reward $r$ and $\mathcal{T}_h^r \mathcal{Q}_{h+1} = \{\mathcal{T}_h^r Q_{h+1} : Q_{h+1} \in \mathcal{Q}_{h+1}\}$.*

In short, Assumption 2 states that the Q-value class $\mathcal{Q}$ should capture the optimal Q-value function, while Assumption 3 indicates the closeness of $\mathcal{Q}$ under Bellman update. It is easy to verify that Assumptions 1, 2 and 3 are more general than the tabular MDP [45, 64], linear mixture MDP [33] and linear MDP [57] assumptions used in previous works.

When the function class contains a finite number of elements, its cardinality can be used to quantify its "size". However, for general function approximation, where the function class may contain an infinite number of elements, we utilize the standard $\varepsilon$-covering number [58] to measure its complexity.

**Definition 1** ($\varepsilon$-covering number). *For function class $\mathcal{F} \subseteq (\mathcal{X} \to \mathbb{R})$, the $\varepsilon$-covering number of $\mathcal{F}$, denoted as $\mathcal{N}_\varepsilon(\mathcal{F})$, is defined as the minimum integer $n$ such that there exists a finite subset $\mathcal{F}' \subseteq \mathcal{F}$ with $|\mathcal{F}'| = n$ such that for any function $f \in \mathcal{F}$, there exists $f' \in \mathcal{F}'$ satisfying that $\max_{x \in \mathcal{X}} |f(x) - f'(x)| \leq \varepsilon$.*

# 4 Optimization-based Adversarial Imitation Learning

In this section, we introduce a provably efficient method called Optimization-Based Adversarial Imitation Learning (OPT-AIL). In Section 4.1, we delve into the core components of OPT-AIL, which involves online optimization for reward functions and optimism-regularized Bellman error minimization for Q-value functions. We discuss the underlying principles and provide theoretical guarantees with general function approximation. Thanks to its easy-to-implement merit, we provide a practical implementation of OPT-AIL using stochastic-gradient-based methods in Section 4.2.

## 4.1 Theoretical Analysis of OPT-AIL

In this section, we present our provably efficient method OPT-AIL with general function approximation; see Algorithm 1 for an overview. To start with, recall that our theoretical goal is to ensure the algorithm can output a policy with $\varepsilon$-imitation gap by using finite expert samples and environment interactions. To obtain the final policy, we leverage the standard online-to-batch conversion technique [37]. Specifically, during the learning process, the learner iteratively generates a sequence of rewards $\{r^k\}_{k=1}^K$ and policies $\{\pi^k\}_{k=1}^K$, and outputs the policy $\overline{\pi}$ that is uniformly sampled from $\{\pi^k\}_{k=1}^K$. To analyze the imitation gap of $\overline{\pi}$, we leverage the following standard error decomposition lemma.

**Lemma 1.** *Consider a sequence of rewards $\{r^k\}_{k=1}^K$ and policies $\{\pi^k\}_{k=1}^K$, and the policy $\overline{\pi}$ is uniformly sampled from $\{\pi^k\}_{k=1}^K$. Then it holds that*

$$V^{\pi^{\mathrm{E}}} - V^{\overline{\pi}} = \underbrace{\frac{1}{K}\sum_{k=1}^K \left(V_{r^{\mathrm{true}}}^{\pi^{\mathrm{E}}} - V_{r^{\mathrm{true}}}^{\pi^k} - \left(V_{r^k}^{\pi^{\mathrm{E}}} - V_{r^k}^{\pi^k}\right)\right)}_{reward\ error} + \underbrace{\frac{1}{K}\sum_{k=1}^K \left(V_{r^k}^{\pi^{\mathrm{E}}} - V_{r^k}^{\pi^k}\right)}_{policy\ error}. \quad (1)$$

Lemma 1 suggests that to achieve a small imitation gap, it is crucial to control both reward error and policy error. Specifically, reward error quantifies the distance between the true reward $r^{\mathrm{true}}$ and the learned reward $r^k$ through the imitation gap. Besides, policy error measures the value difference between the expert policy $\pi^{\mathrm{E}}$ and the learned policy $\pi^k$ under the inferred reward $r^k$. Notably, this policy error differs from the concept of regret in RL [23, 32], where the reward is fixed.

To theoretically minimize the reward error and policy error, we consider an iterative approach, in which each iteration first updates the reward and subsequently derives the policy. The subsequent parts detail the reward and policy updates, which involve solving two optimization problems.

---

**Algorithm 1** Optimization-based Adversarial Imitation Learning

**Input:** Reward class $\mathcal{R}$, Q-value class $\mathcal{Q}$, initialized reward $r^0$, policy $\pi^0$ and dataset $\mathcal{D}^0 = \emptyset$.
1: **for** $k = 1, 2, \ldots, K$ **do**
2:      Apply $\pi^{k-1}$ to roll out a trajectory $\tau^{k-1}$ and append it to the dataset $\mathcal{D}^k = \mathcal{D}^{k-1} \cup \{\tau^{k-1}\}$.
3:      Obtain $r^k$ by running a no-regret algorithm to solve the online optimization problem with observed loss functions $\{\mathcal{L}^i(r)\}_{i=0}^{k-1}$ up to an error $\varepsilon_{\mathrm{opt}}^r$, where $\mathcal{L}^i(r) = \widehat{V}_r^{\pi^i} - \widehat{V}_r^{\pi^{\mathrm{E}}}$.
4:      Obtain $Q^k$ by solving the following optimization problem up to an error $\varepsilon_{\mathrm{opt}}^Q$.

$$\min_{Q \in \mathcal{Q}} \mathcal{L}^k(Q) := \mathrm{BE}^k(Q) - \lambda \max_{a \in \mathcal{A}} Q_1(s_1, a),$$

     where $\mathrm{BE}^k(Q) = \sum_{h=1}^H \mathcal{E}_h(Q_h, Q_{h+1}; \mathcal{D}^k, r^k) - \inf_{Q_h' \in \mathcal{Q}_h} \mathcal{E}_h(Q_h', Q_{h+1}; \mathcal{D}^k, r^k)$.
5:      Obtain $\pi^k$ by $\pi_h^k(s) = \mathrm{argmax}_{a \in \mathcal{A}} Q_h^k(s, a)$.
6: **end for**
**Output:** $\overline{\pi}$ sampled uniformly from $\{\pi^k\}_{k=1}^K$.

---

**Reward Update via Online Optimization (Line 3 in Algorithm 1).** The goal of this step is to control the reward error. More concretely, in iteration $k$, we aim to learn a reward $r^k$ such that the error $V_{r^k}^{\pi^k} - V_{r^k}^{\pi^{\mathrm{E}}} - (V_{r^{\mathrm{true}}}^{\pi^k} - V_{r^{\mathrm{true}}}^{\pi^{\mathrm{E}}})$ is small. We formulate this problem as an *online* optimization problem. In iteration $k$, using the previously observed loss functions $\{V_r^{\pi^i} - V_r^{\pi^{\mathrm{E}}}\}_{i=0}^{k-1}$, the reward learner selects $r^k$ and then observes the current loss function $V_r^{\pi^k} - V_r^{\pi^{\mathrm{E}}}$. Moreover, since the previous expected loss functions $\{V_r^{\pi^i} - V_r^{\pi^{\mathrm{E}}}\}_{i=0}^{k-1}$ are not available, we instead minimize the *estimated* loss

functions. In particular, we leverage expert demonstrations $\mathcal{D}^{\mathrm{E}}$ and the trajectory $\tau^i$ collected by policy $\pi^i$ to establish an unbiased estimation $\mathcal{L}^i(r) = \widehat{V}_r^{\pi^i} - \widehat{V}_r^{\pi^{\mathrm{E}}}$ for $V_r^{\pi^i} - V_r^{\pi^{\mathrm{E}}}$, where

$$\widehat{V}_r^{\pi^i} = \sum_{h=1}^{H} r_h(s_h^i, a_h^i), \ \widehat{V}_r^{\pi^{\mathrm{E}}} = \frac{1}{N} \sum_{\tau \in \mathcal{D}^{\mathrm{E}}} \sum_{h=1}^{H} r_h(\tau(s_h), \tau(a_h)).$$

Here $(\tau(s_h), \tau(a_h))$ is the state-action pair of trajectory $\tau$ visited in time step $h$ and $\tau^i = \{s_1^i, a_1^i, \ldots, s_H^i, a_H^i\}$ is the trajectory collected by policy $\pi^i$. The ultimate goal of the reward learner is to minimize the cumulative losses $\sum_{k=1}^{K} \widehat{V}_{r^k}^{\pi^k} - \widehat{V}_{r^k}^{\pi^{\mathrm{E}}}$. To achieve this goal, we employ a no-regret algorithm [18]. In the following part, we formally define the reward optimization error resulting from running the no-regret algorithm.

**Definition 2** (Reward Optimization Error). *For any sequence of policies $\{\pi^k\}_{k=1}^{K}$, the no-regret reward optimization algorithm sequentially outputs rewards $r^1, \ldots, r^K$. The reward optimization error $\varepsilon_{\mathrm{opt}}^r$ is defined as $\varepsilon_{\mathrm{opt}}^r := (1/K) \cdot \max_{r \in \mathcal{R}} \sum_{k=1}^{K} \widehat{V}_{r^k}^{\pi^k} - \widehat{V}_{r^k}^{\pi^{\mathrm{E}}} - (\widehat{V}_r^{\pi^k} - \widehat{V}_r^{\pi^{\mathrm{E}}}).$*

The reward optimization error, as defined above, aligns with the standard average regret in online optimization [18], a concept not extensively explored in the context of AIL. When the loss functions $\{\mathcal{L}^k(r)\}_{k=0}^{K}$ are convex functions and the reward class $\mathcal{R}$ is a convex set, we can apply online projected gradient descent [18] as the no-regret algorithm, which ensures the reward optimization error $\varepsilon_{\mathrm{opt}}^r = \mathcal{O}(1/\sqrt{K})$. As for non-convex functions and sets, employing Follow-the-Perturbed-Leader can similarly achieve $\varepsilon_{\mathrm{opt}}^r = \mathcal{O}(1/\sqrt{K})$ [47].

**Policy Update via Optimism-Regularized Bellman-error Minimization (Lines 4-5 in Algorithm 1).** The target of policy updates is to control the policy error. In iteration $k$, the policy learner aims to recover a policy $\pi^k$ such that the policy error $V_{r^k}^{\pi^{\mathrm{E}}} - V_{r^k}^{\pi^k}$ is small, where $r^k$ is the recently recovered reward function. This is essentially an RL problem under reward function $r^k$. Building upon [32], we leverage a model-free approach, based on Bellman error minimization, to solve this RL sub-problem. In particular, we first learn Q-value functions by solving the optimization problem of

$$\min_{Q \in \mathcal{Q}} \mathcal{L}^k(Q) := \mathrm{BE}^k(Q) - \lambda \max_{a \in \mathcal{A}} Q_1(s_1, a),$$

$$\text{with } \mathrm{BE}^k(Q) = \sum_{h=1}^{H} \mathcal{E}_h(Q_h, Q_{h+1}; \mathcal{D}^k, r^k) - \inf_{Q_h' \in \mathcal{Q}_h} \mathcal{E}_h(Q_h', Q_{h+1}; \mathcal{D}^k, r^k),$$

where $\mathcal{E}_h(Q_h, Q_{h+1}; \mathcal{D}^k, r^k) = \sum_{i=0}^{k-1}(Q_h(s_h^i, a_h^i) - r_h^k - \max_{a' \in \mathcal{A}} Q_{h+1}(s_{h+1}^i, a'))^2$, $\mathcal{D}^k = \{\tau^i\}_{i=0}^{k-1}$ with $\tau^i = \{s_1^i, a_1^i, \ldots, s_H^i, a_H^i\}$ and $\lambda > 0$ is the regularization coefficient. As shown in [5, 23], $\mathrm{BE}^k(Q)$ is the estimated squared Bellman error of $Q$ with respect to reward $r^k$ and dataset $\mathcal{D}^k$. In this optimization problem, the main objective $\mathrm{BE}^k(Q)$ ensures a small Bellman error while the regularization term $\max_{a \in \mathcal{A}} Q_1(s_1, a)$ tends to search an optimistic Q-value function for encouraging exploration. It is worth noting that Algorithm 1 only requires approximately solving the optimization problem up to an error $\varepsilon_{\mathrm{opt}}^Q$ with $\varepsilon_{\mathrm{opt}}^Q = \mathcal{L}^k(Q^k) - \min_{Q \in \mathcal{Q}} \mathcal{L}^k(Q)$. After obtaining the Q-value function $Q^k$, we derive $\pi^k$ as the greedy policy of $Q^k$.

**Theoretical Guarantee of OPT-AIL.** In the above part, we have explained the algorithmic mechanisms of OPT-AIL. Now we present the theoretical guarantee. To ensure the sample efficiency of solving RL sub-problems within AIL, we make a structural assumption on the underlying MDP. In particular, we assume that the MDP has a small generalized eluder coefficient. This coefficient, introduced in [68], quantifies the inherent difficulty of learning the MDP with function approximation in RL. We adapt this concept to AIL where the reward function is changing.

**Assumption 4** (Low generalized eluder coefficient [68]). *We assume that given an $\varepsilon > 0$, the generalized eluder coefficient $d_{\mathrm{GEC}}(\varepsilon)$ is the smallest $d$ $(d \geq 0)$ such that for any sequence of $\{r^k\}_{k=1}^{K} \subseteq \mathcal{R}$, $\{Q^k\}_{k=1}^{K} \subseteq \mathcal{Q}$ and the corresponding greedy policies $\{\pi^k\}_{k=1}^{K}$,*

$$\sum_{k=1}^{K} Q_1^k(s_1, \pi^k) - V_{r^k}^{\pi^k} \leq \inf_{\mu \geq 0} \frac{\mu}{2} \sum_{k=1}^{K} \sum_{i=1}^{k-1} \mathbb{E}\left[\sum_{h=1}^{H}\left(Q_h^k(s_h, a_h) - \mathcal{T}_h^{r^k} Q_{h+1}^k(s_h, a_h)\right)^2 \bigg| \pi^i\right] + \frac{d}{2\mu}$$

$$+ \sqrt{dHK} + \varepsilon HK,$$

*where $Q_1^k(s_1, \pi^k) := \mathbb{E}_{a_1 \sim \pi_1^k(\cdot | s_1)}[Q_1^k(s_1, a_1)]$.*

Intuitively, a low generalized eluder coefficient ensures that the prediction error $Q_1^k(s_1, \pi^k) - V_{r^k}^{\pi^k}$ for $\pi^k$ can be effectively controlled by the Bellman error on the dataset generated by historical policies $\{\pi^i\}_{i=1}^{k-1}$. As demonstrated in [68], the MDPs with low generalized eluder coefficient form a rich class of MDPs, which covers many well-known MDP instances such tabular MDPs, linear MDPs [24] and MDPs with low Bellman eluder dimension [23]. Now we are ready to present the theoretical guarantee of OPT-AIL.

**Theorem 1.** *Under Assumptions 1, 2, 3 and 4. For any fixed $\varepsilon \in (0, 1]$ and $\delta \in (0, 1]$, consider Algorithm 1 with $\lambda = c_1\sqrt{(KH^3\log(4KHN_\rho(\mathcal{Q})N_\rho(\mathcal{R})/\delta) + K^2H^3\rho)/d_{\mathrm{GEC}}}$, where $d_{\mathrm{GEC}} := d_{\mathrm{GEC}}(\varepsilon/H)$, $\rho := c_2\varepsilon^2/(H^2d_{\mathrm{GEC}} + H)$, $c_1$ and $c_2$ are absolute constants. Then with probability at least $1 - \delta$, we have that $V^{\pi^{\mathrm{E}}} - V^{\bar\pi} \leq \varepsilon + \varepsilon_{\mathrm{opt}}^r + (\varepsilon_{\mathrm{opt}}^Q/\lambda)$ if the expert sample complexity and interaction complexity satisfy*

$$N \gtrsim \frac{H^2\log(\max_{h\in[H]}\mathcal{N}_\rho(\mathcal{R}_h)/\delta)}{\varepsilon^2},$$

$$K \gtrsim \frac{H^4d_{\mathrm{GEC}}\log(Hd_{\mathrm{GEC}}\max_{h\in[H]}\mathcal{N}_\rho(\mathcal{Q}_h)\mathcal{N}_\rho(\mathcal{R}_h)/(\delta\varepsilon)) + H^2\log(1/\delta)}{\varepsilon^2}.$$

The proof of Theorem 1 can be found in Appendix B.2. Theorem 1 indicates that when $d_{\mathrm{GEC}} = \Omega(1)$, OPT-AIL achieves the expert sample complexity $\widetilde{\mathcal{O}}(H^2\log(\max_{h\in[H]}\mathcal{N}_\rho(\mathcal{R}_h))/\varepsilon^2)$ and interaction complexity $\widetilde{\mathcal{O}}(H^4d_{\mathrm{GEC}}\log(\max_{h\in[H]}\mathcal{N}_\rho(\mathcal{Q}_h)\mathcal{N}_\rho(\mathcal{R}_h))/\varepsilon^2)$ in the general function approximation setting. To our best knowledge, OPT-AIL is the first provably efficient online AIL algorithm for general function approximation.

Notably, OPT-AIL improves over BC [13] by an order of $\mathcal{O}(H)$, suggesting that OPT-AIL can still provably mitigate the compounding errors issue in BC for general function approximation. When restricting Theorem 1 to linear MDPs with dimension $d$ [68], OPT-AIL can achieve the expert sample complexity $\widetilde{\mathcal{O}}(H^2d/\varepsilon^2)$ and interaction complexity $\widetilde{\mathcal{O}}(H^5d^2/\varepsilon^2)$. Furthermore, when $\mathcal{Q}$ and $\mathcal{R}$ are neural network classes commonly employed in practice, we can obtain the corresponding complexity result by plugging the covering number bound of neural networks [7] into Theorem 1. Finally, OPT-AIL only requires the approximate optimization of two objectives, thereby facilitating a practical implementation with neural networks, which will be presented in the next section.

Although Theorem 1 produces desirable outcomes, it does have some limitations. One of the limitations is that Theorem 1 requires the Bellman completeness condition for the Q-value class (i.e., Assumption 3). Nevertheless, recent advances [4] in RL have developed new techniques to remove this assumption. We leave the extension of these techniques to AIL for future work.

### 4.2 Practical Implementation of OPT-AIL

In this section, we provide a practical implementation for OPT-AIL, which is based on the stochastic-gradient-based methods; see Algorithm 2 for an overview. We elaborate on the practical reward update and policy update in detail as follows.

---
**Algorithm 2** Practical Implementation of OPT-AIL
---

**Input:** Initialized reward $r^0$, Q-value $Q^0$, target Q-value $\overline{Q}^0 = Q^0$, policy $\pi^0$ and dataset $\mathcal{D}^0 = \emptyset$.
 1: **for** $k = 1, 2, \ldots, K$ **do**
 2:     Apply $\pi^{k-1}$ to roll out a trajectory $\tau^{k-1}$ and append it to the dataset $\mathcal{D}^k = \mathcal{D}^{k-1} \cup \{\tau^{k-1}\}$.
 3:     Update the reward function by $r^k \leftarrow r^{k-1} - \alpha_r\nabla\ell^k(r)$ from Eq. (2).
 4:     Update the Q-value function by $Q^k \leftarrow Q^{k-1} - \alpha_Q\nabla\ell^k(Q)$ from Eq. (3).
 5:     Update the policy by $\pi^k \leftarrow \pi^{k-1} + \alpha_\pi\nabla\ell^k(\pi)$, where $\ell^k(\pi) := \mathbb{E}_{\tau\sim\mathcal{D}^k}[\sum_{h=1}^H Q_h^k(s_h, \pi)]$.
 6:     Update the target Q-value by $\overline{Q}^k \leftarrow \tau Q^k + (1 - \tau)\overline{Q}^{k-1}$
 7: **end for**

---

**Practical Reward Update.** Here we detail a practical implementation of the reward update by applying an online optimization approach. Recall that in line 3 of Algorithm 1, a no-regret algorithm is employed to solve the online optimization problem. To implement this mechanism, we choose the classical online optimization approach Follow-the-Regularized-Leader (FTRL) [2] as the no-regret

approach. In iteration $k$, FTRL minimizes the sum of all historical loss functions with a regularization.

$$
\begin{aligned}
\min_{r \in \mathcal{R}} \ell^k(r) :&= \sum_{i=0}^{k-1} \mathcal{L}^i(r) + \beta\psi(r) \\
&= k\left( \mathbb{E}_{\tau \sim \mathcal{D}^k}\left[ \sum_{h=1}^{H} r_h(s_h^i, a_h^i) \right] - \mathbb{E}_{\tau \sim \mathcal{D}^{\mathrm{E}}}\left[ \sum_{h=1}^{H} r_h(s_h^i, a_h^i) \right] \right) + \beta\psi(r),
\end{aligned}
\tag{2}
$$

where $\mathbb{E}_{\mathcal{D}}[\cdot]$ denotes the empirical distribution of dataset $\mathcal{D}$. Here $\psi(r)$ is the regularization term. In practice, we choose $\psi(r)$ as the gradient penalty [6] of the reward model, which can help stabilize the learning process [27]. According to Eq. (2), the reward learner aims to maximize the value gap between the expert policy and all previous policies.

Besides, as indicated in Eq. (2), all historical samples $\mathcal{D}^k$ are utilized for the reward update. This learning style is exactly off-policy reward learning [27, 28]. In particular, applying FTRL for the reward update and off-policy reward learning share the same main objective. Previous practical works [27, 28] found that this off-policy reward learning works well in practice, but could not give an explanation. In this work, we justify this off-policy learning style from an online optimization perspective: performing off-policy learning, which aligns with the FTRL approach, can effectively control the reward optimization error.

**Practical Policy Update.** To implement the policy update in practice, we adopt the actor-critic framework [14, 17, 27]. In particular, we maintain a policy model $\pi$ and a Q-value model $Q$. Recall in line 4 of Algorithm 1, the Q-value function is learned by minimizing the optimism-regularized Bellman error. To implement this principle, following [12, 8], we leverage the temporal difference loss [31] of the Q-value model and its delayed target to approximate the theoretical Bellman error. Then we arrive at the following objective.

$$
\min_{Q \in \mathcal{Q}} \ell^k(Q) := \mathbb{E}_{\tau \sim \mathcal{D}^k}\left[ \sum_{h=1}^{H} \left( Q_h(s_h, a_h) - r_h^k - \overline{Q}_{h+1}^{k-1}(s_{h+1}, \pi^{k-1}) \right)^2 \right] - \lambda Q_1(s_1, \pi^{k-1}). \tag{3}
$$

Here $\overline{Q} = \{\overline{Q}_1, \dots, \overline{Q}_H\}$ is the delayed target Q-value model. Besides, we define that $\overline{Q}_{h+1}^{k-1}(s_{h+1}, \pi^{k-1}) := \mathbb{E}_{a' \sim \pi_{h+1}^{k-1}(\cdot|s_{h+1})}[\overline{Q}_{h+1}(s_{h+1}, a')]$ where the previous greedy policy $\pi^{k-1}$ is used to approximate the maximum operator [17]. Consequently, we derive the greedy policy by optimizing the objective of $\max_\pi \ell^k(\pi) := \mathbb{E}_{\tau \sim \mathcal{D}^k}[\sum_{h=1}^{H} Q_h^k(s_h, \pi)]$.

# 5  Experiments

In this section, we evaluate the expert sample efficiency and environment interaction efficiency of OPT-AIL through experiments. Below, we provide a brief overview of the experimental set-up, with detailed information available in Appendix C due to space constraints.

## 5.1  Experiment Set-up

**Environment.** We conduct experiments on 8 tasks sourced from the feature-based DMControl benchmark [53], a leading benchmark in IL that offers a diverse set of continuous control tasks. For each task, we adopt online DrQ-v2 [66] to train an agent with sufficient environment interactions and regard the resultant policy as the expert policy. Then we roll out this expert policy to collect expert demonstrations. Each algorithm is tested over five trials with different random seeds, and in each run, we evaluate the policy return using Monte Carlo approximation with 10 trajectories.

**Baselines.** Existing theoretical AIL approaches like MB-TAIL [64] and OGAIL [33] include count-based or covariance-based bonuses, making it challenging to implement these designs when using neural network approximations. Thus, we do not include these methods in our experiments. Instead, we compare OPT-AIL with prior deep IL methods, including BC [39], IQLearn [15], PPIL [56], FILTER [51] and HyPE [42], despite that most of them lack theoretical guarantees. Notably, IQLearn, FILTER and HyPE represent prior SOTA deep AIL approaches. To ensure a fair comparison, we implement all these methods within the same codebase. For detailed implementations, please refer to Appendix C.

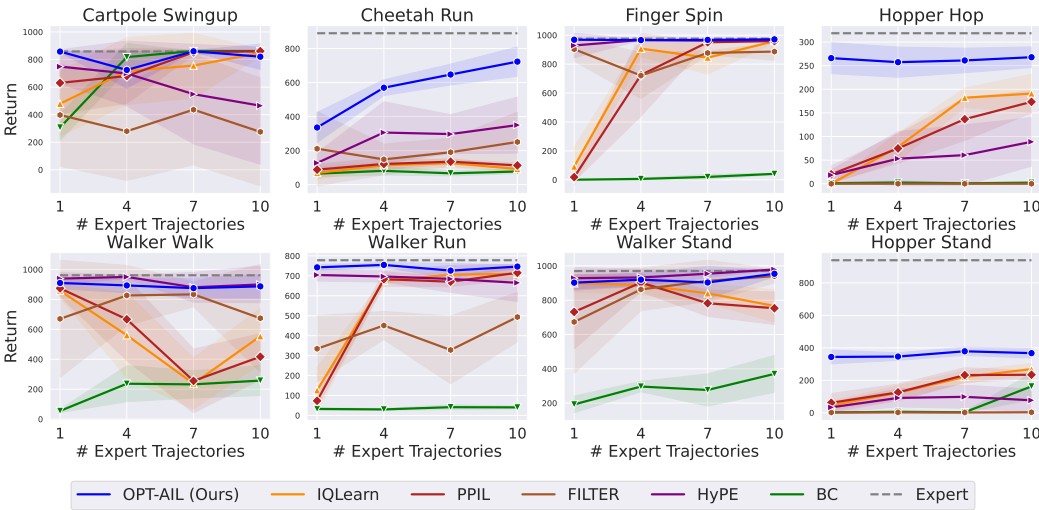

Figure 1: Overall performance on 8 DMControl tasks over 5 random seeds following 500k interactions with the environment. Here the $x$-axis is the number of expert trajectories and the $y$-axis is the return. The solid lines are the mean of results while the shaded region corresponds to the standard deviation over 5 random seeds. Same as the following figures.

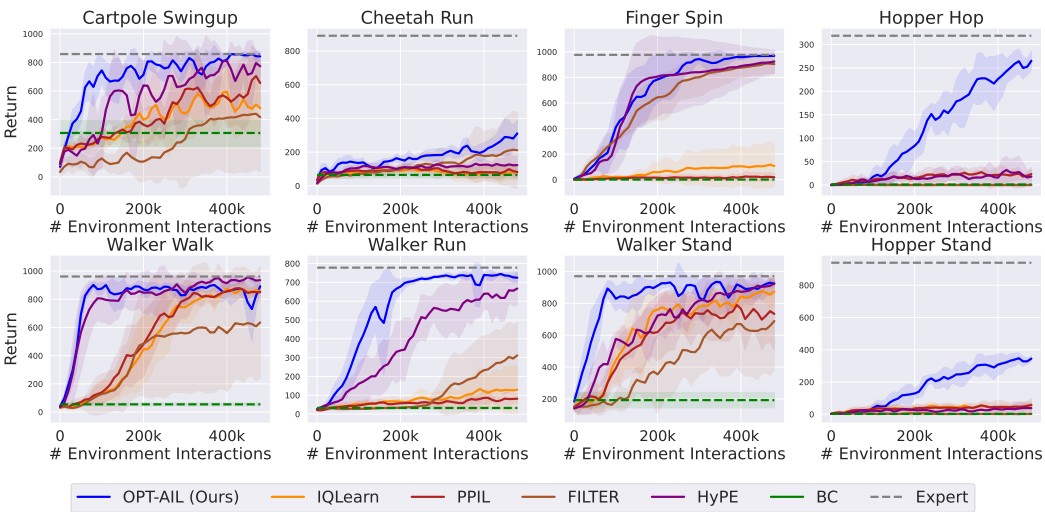

Figure 2: Learning curves on 8 DMControl tasks over 5 random seeds using 1 expert trajectory. Here the $x$-axis is the number of environment interactions and the $y$-axis is the return.

## 5.2 Experiment Results

**Expert Sample Efficiency.** Figure 1 shows the performance of the learned policies after 500k environment interactions with varying numbers of expert trajectories. First, OPT-AIL significantly outperforms BC, verifying the theoretical claim that OPT-AIL can mitigate the compounding errors issue inherent in BC for general function approximation. Moreover, OPT-AIL consistently matches or exceeds the performance of prior SOTA AIL methods on all tasks. Notably, OPT-AIL demonstrates outstanding performance in scenarios with limited expert demonstrations, a common occurrence in real-world applications. In particular, when there is only one expert trajectory, our method uniquely achieves expert or near-expert performance on tasks like `Finger Spin`, `Walker Run` and `Hopper Hop`.

**Environment Interaction Efficiency.** Figure 2 displays the learning curves of different algorithms with 1 expert trajectory. Compared with prior SOTA AIL approaches, OPT-AIL achieves comparable or better performance regarding interaction efficiency on all 8 tasks. Notably, on `Hopper Hop`, `Walker Run` and `Walker Run`, OPT-AIL can achieve near-expert performance with substantially fewer environment interactions compared with prior approaches. We also demonstrate the superior interaction efficiency of OPT-AIL with other numbers of expert trajectories; please refer to Appendix D for additional results.

## 6  Conclusions

To narrow the gap between theory and practice in adversarial imitation learning, this paper investigates AIL with general function approximation. We develop a new AIL approach termed OPT-AIL, which centers on performing online optimization for reward functions and optimism-regularized Bellman error minimization for Q-value functions. In theory, OPT-AIL achieves polynomial expert sample complexity and interaction complexity for general function approximation. In practice, OPT-AIL only requires approximately solving two optimization problems, which enables an efficient implementation. Our experiments demonstrate that OPT-AIL outperforms prior SOTA methods in several challenging tasks, highlighting its potential to bridge theoretical soundness with practical efficiency.

In tabular MDPs, the currently optimal expert sample complexity is $\mathcal{O}(H^{3/2}/\varepsilon)$ [41, 64], which is better than $\mathcal{O}(H^2/\varepsilon^2)$ attained in this paper. Therefore, a promising and valuable future direction would be to develop more advanced AIL approaches that achieve this expert sample complexity in the setting of general function approximation. Besides, [63] established a horizon-free imitation gap bound for AIL in tabular MDPs. Thus it is interesting to explore horizon-free bounds for AIL with general function approximation.

## 7  Acknowledgements

We thank Ziniu Li and Yichen Li for their helpful discussions and feedback. This work was supported by the Fundamental Research Program for Young Scholars (PhD Candidates) of the National Science Foundation of China (623B2049) and Jiangsu Science Foundation (BK20243039).

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

## A  Broader Impacts

This study explores the theoretical foundations of adversarial imitation learning with general function approximation and demonstrates the efficiency of the proposed algorithm through standard benchmarks. Although the paper does not reveal any immediate social impacts, the potential practical applications of our research could drive positive change. By broadening the scope of adversarial imitation learning, our work may enable the creation of more efficient and effective solutions in fields such as robotics and autonomous vehicles. Nonetheless, we must recognize the potential for negative consequences if this technology is misused. For example, imitation learning learns from human expert demonstrations and could raise privacy concerns. Therefore, it is essential to ensure that the advancements in imitation learning are applied responsibly and ethically.

## B  Omitted Proof

### B.1  Proof of Lemma 1

Lemma 1 is a standard error decomposition lemma in adversarial imitation learning and variants of Lemma 1 have appeared in [45, 57]. According to the definition of $\overline{\pi}$, we have that

$$
\begin{aligned}
V^{\pi^{\mathrm{E}}} - V^{\overline{\pi}} &= V_{r^{\mathrm{true}}}^{\pi^{\mathrm{E}}} - V_{r^{\mathrm{true}}}^{\overline{\pi}} \\
&= \frac{1}{K}\sum_{k=1}^{K} V_{r^{\mathrm{true}}}^{\pi^{\mathrm{E}}} - V_{r^{\mathrm{true}}}^{\pi^{k}} \\
&= \frac{1}{K}\sum_{k=1}^{K}\left(V_{r^{\mathrm{true}}}^{\pi^{\mathrm{E}}} - V_{r^{\mathrm{true}}}^{\pi^{k}} - \left(V_{r^{k}}^{\pi^{\mathrm{E}}} - V_{r^{k}}^{\pi^{k}}\right)\right) + \frac{1}{K}\sum_{k=1}^{K} V_{r^{k}}^{\pi^{\mathrm{E}}} - V_{r^{k}}^{\pi^{k}}.
\end{aligned}
$$

We complete the proof.

### B.2  Proof of Theorem 1

In this section, we present the proof of Theorem 1.

To prove Theorem 1, we need the following two useful lemmas which upper bound the reward error and policy error, respectively. Please refer to Appendix B.3 and B.4 for the detailed proof.

**Lemma 2** (Upper Bound on Reward Error). *Under Assumption 1. For any fixed $\delta \in (0,1]$, consider Algorithm 1, with probability at least $1-\delta$,*

$$
\frac{1}{K}\sum_{k=1}^{K} V_{r^{\mathrm{true}}}^{\pi^{\mathrm{E}}} - V_{r^{\mathrm{true}}}^{\pi^{k}} - \left(V_{r^{k}}^{\pi^{\mathrm{E}}} - V_{r^{k}}^{\pi^{k}}\right) \le 2H\sqrt{\frac{\log(6\max_{h\in[H]}\mathcal{N}_\rho(\mathcal{R}_h)/\delta)}{N}} + 4H\rho
$$
$$
+ 2H\sqrt{\frac{\log(3/\delta)}{K}} + \varepsilon_{\mathrm{opt}}^{r}.
$$

**Lemma 3** (Upper Bound on Policy Error). *Under Assumptions 2, 3 and 4. For any fixed $\delta \in (0,1]$, with probability at least $1-\delta$, it holds that*

$$
\frac{1}{K}\sum_{k=1}^{K} V_{r^{k}}^{\pi^{\mathrm{E}}} - V_{r^{k}}^{\pi^{k}} \le \frac{57H^4\log(4KH\max_{h\in[H]}\mathcal{N}_\rho(\mathcal{Q}_h)\mathcal{N}_\rho(\mathcal{R}_h)/\delta) + 57KH^3\rho + \varepsilon_{\mathrm{opt}}^{Q}}{\lambda}
$$
$$
+ \frac{\lambda d_{\mathrm{GEC}}(\varepsilon')}{2K} + \sqrt{\frac{d_{\mathrm{GEC}}(\varepsilon')H}{K}} + \varepsilon' H.
$$

Now we start to prove Theorem 1. With Lemma 1, we can derive that

$$
V_{r^{\mathrm{true}}}^{\pi^{\mathrm{E}}} - V_{r^{\mathrm{true}}}^{\overline{\pi}} = \frac{1}{K}\sum_{k=1}^{K} V_{r^{\mathrm{true}}}^{\pi^{\mathrm{E}}} - V_{r^{\mathrm{true}}}^{\pi^{k}} - \left(V_{r^{k}}^{\pi^{\mathrm{E}}} - V_{r^{k}}^{\pi^{k}}\right) + \frac{1}{K}\sum_{k=1}^{K} V_{r^{k}}^{\pi^{\mathrm{E}}} - V_{r^{k}}^{\pi^{k}}.
$$

Furthermore, Lemma 2 and Lemma 3 offer upper bounds on reward error and policy error, respectively. By union bound, with probability at least $1 - \delta$, we obtain

$$V_{r^{\text{true}}}^{\pi^{\text{E}}} - V_{r^{\text{true}}}^{\overline{\pi}}$$

$$\leq 2H\sqrt{\frac{\log(12\max_{h\in[H]}\mathcal{N}_\rho(\mathcal{R}_h)/\delta)}{N}} + 4H\rho + 2H\sqrt{\frac{\log(6/\delta)}{K}} + \varepsilon_{\text{opt}}^r$$

$$+ \frac{57H^4\log(8KH\max_{h\in[H]}\mathcal{N}_\rho(\mathcal{Q}_h)\mathcal{N}_\rho(\mathcal{R}_h)/\delta) + 57KH^3\rho + \varepsilon_{\text{opt}}^Q}{\lambda}$$

$$+ \frac{\lambda d_{\text{GEC}}(\varepsilon')}{2K} + \sqrt{\frac{d_{\text{GEC}}(\varepsilon')H}{K}} + \varepsilon'H.$$

We choose $\varepsilon' = \varepsilon/H$ and obtain

$$V_{r^{\text{true}}}^{\pi^{\text{E}}} - V_{r^{\text{true}}}^{\overline{\pi}}$$

$$\leq 2H\sqrt{\frac{\log(12\max_{h\in[H]}\mathcal{N}_\rho(\mathcal{R}_h)/\delta)}{N}} + 4H\rho + 2H\sqrt{\frac{\log(6/\delta)}{K}}$$

$$+ \frac{57H^4\log(8KH\max_{h\in[H]}\mathcal{N}_\rho(\mathcal{Q}_h)\mathcal{N}_\rho(\mathcal{R}_h)/\delta) + 57KH^3\rho}{\lambda}$$

$$+ \frac{\lambda d_{\text{GEC}}}{2K} + \sqrt{\frac{d_{\text{GEC}}H}{K}} + \varepsilon_{\text{opt}}^r + \frac{\varepsilon_{\text{opt}}^Q}{\lambda} + \varepsilon,$$

where $d_{\text{GEC}} := d_{\text{GEC}}(\varepsilon/H)$. By choosing the regularization coefficient

$$\lambda = \sqrt{\frac{114KH^4\log(8KH\max_{h\in[H]}\mathcal{N}_\rho(\mathcal{Q}_h)\mathcal{N}_\rho(\mathcal{R}_h)/\delta) + 114K^2H^3\rho}{d_{\text{GEC}}}},$$

we further obtain

$$V_{r^{\text{true}}}^{\pi^{\text{E}}} - V_{r^{\text{true}}}^{\overline{\pi}}$$

$$\leq 2H\sqrt{\frac{\log(12\max_{h\in[H]}\mathcal{N}_\rho(\mathcal{R}_h)/\delta)}{N}} + 4H\rho + 2H\sqrt{\frac{\log(6/\delta)}{K}}$$

$$+ \sqrt{\frac{114H^4 d_{\text{GEC}}\log(8KH\max_{h\in[H]}\mathcal{N}_\rho(\mathcal{Q}_h)\mathcal{N}_\rho(\mathcal{R}_h)/\delta)}{K} + 114H^3 d_{\text{GEC}}\rho} + \sqrt{\frac{d_{\text{GEC}}H}{K}}$$

$$+ \varepsilon_{\text{opt}}^r + \frac{\varepsilon_{\text{opt}}^Q}{\lambda} + \varepsilon$$

$$\overset{(a)}{\leq} 2H\sqrt{\frac{\log(12\max_{h\in[H]}\mathcal{N}_\rho(\mathcal{R}_h)/\delta)}{N}} + 4H\rho + 2H\sqrt{\frac{\log(6/\delta)}{K}}$$

$$+ \sqrt{\frac{114H^4 d_{\text{GEC}}\log(8KH\max_{h\in[H]}\mathcal{N}_\rho(\mathcal{Q}_h)\mathcal{N}_\rho(\mathcal{R}_h)/\delta)}{K}} + \sqrt{114H^3 d_{\text{GEC}}\rho} + \sqrt{\frac{d_{\text{GEC}}H}{K}}$$

$$+ \varepsilon_{\text{opt}}^r + \frac{\varepsilon_{\text{opt}}^Q}{\lambda} + \varepsilon$$

$$\leq 2H\sqrt{\frac{\log(12\max_{h\in[H]}\mathcal{N}_\rho(\mathcal{R}_h)/\delta)}{N}} + 4H\rho + 2H\sqrt{\frac{\log(6/\delta)}{K}}$$

$$+ 2\sqrt{\frac{114H^4 d_{\text{GEC}}\log(8KH\max_{h\in[H]}\mathcal{N}_\rho(\mathcal{Q}_h)\mathcal{N}_\rho(\mathcal{R}_h)/\delta)}{K}} + \sqrt{54H^3 d_{\text{GEC}}\rho}$$

$$+ \varepsilon_{\text{opt}}^r + \frac{\varepsilon_{\text{opt}}^Q}{\lambda} + \varepsilon$$

$$\overset{(b)}{\leq} 2H\sqrt{\frac{\log(12\max_{h\in[H]}\mathcal{N}_\rho(\mathcal{R}_h)/\delta)}{N}} + 2H\sqrt{\frac{\log(6/\delta)}{K}}$$

$$+ 24\sqrt{\frac{H^4 d_{\text{GEC}}\log(8KH\max_{h\in[H]}\mathcal{N}_\rho(\mathcal{Q}_h)\mathcal{N}_\rho(\mathcal{R}_h)/\delta)}{K}} + \varepsilon_{\text{opt}}^r + \frac{\varepsilon_{\text{opt}}^Q}{\lambda} + 3\varepsilon$$

Inequality (a) follows $\sqrt{a+b} \leq \sqrt{a} + \sqrt{b}$, $\forall a, b \geq 0$ and inequality (b) holds because of the choice $\rho = \varepsilon^2/(54H^3 d_{\mathrm{GEC}} + 4H)$. Now we determine the number of expert trajectories and the number of interaction trajectories. With Lemma 8, when the expert sample complexity and interaction complexity satisfies

$$N \geq 4\frac{H^2 \log(12 \max_{h \in [H]} \mathcal{N}_\rho(\mathcal{R}_h)/\delta)}{\varepsilon^2},$$

$$K \geq 2304\frac{(H^4 d_{\mathrm{GEC}} \log(768 H^{5/2} d_{\mathrm{GEC}}^{1/2} \max_{h \in [H]} \mathcal{N}_\rho(\mathcal{Q}_h)\mathcal{N}_\rho(\mathcal{R}_h)/(\delta\varepsilon)) + H^2 \log(6/\delta))}{\varepsilon^2},$$

we have that

$$V_{r^{\mathrm{true}}}^{\pi^{\mathrm{E}}} - V_{r^{\mathrm{true}}}^{\bar{\pi}} \leq 6\varepsilon + \varepsilon_{\mathrm{opt}}^r + \frac{\varepsilon_{\mathrm{opt}}^Q}{\lambda}.$$

Scaling $\varepsilon$ as $\varepsilon/6$ completes the proof.

### B.3 Proof of Lemma 2

To prove Lemma 2, we first perform the following error decomposition.

$$
\frac{1}{K}\sum_{k=1}^{K} V_{r^{\mathrm{true}}}^{\pi^{\mathrm{E}}} - V_{r^{\mathrm{true}}}^{\pi^k} - \left(V_{r^k}^{\pi^{\mathrm{E}}} - V_{r^k}^{\pi^k}\right)
$$

$$
= \frac{1}{K}\sum_{k=1}^{K} \left(\widehat{V}_{r^{\mathrm{true}}}^{\pi^{\mathrm{E}}} - \widehat{V}_{r^{\mathrm{true}}}^{\pi^k} - \left(\widehat{V}_{r^k}^{\pi^{\mathrm{E}}} - \widehat{V}_{r^k}^{\pi^k}\right)\right) + V_{r^{\mathrm{true}}}^{\pi^{\mathrm{E}}} - \widehat{V}_{r^{\mathrm{true}}}^{\pi^{\mathrm{E}}} + \frac{1}{K}\sum_{k=1}^{K} \widehat{V}_{r^k}^{\pi^{\mathrm{E}}} - V_{r^k}^{\pi^{\mathrm{E}}} \quad (4)
$$

$$
+ \frac{1}{K}\sum_{k=1}^{K} \widehat{V}_{r^{\mathrm{true}}}^{\pi^k} - V_{r^{\mathrm{true}}}^{\pi^k} + \frac{1}{K}\sum_{k=1}^{K} V_{r^k}^{\pi^k} - \widehat{V}_{r^k}^{\pi^k}.
$$

Recall that for any reward function $r$, $\widehat{V}_r^{\pi^i}$ and $\widehat{V}_r^{\pi^{\mathrm{E}}}$ are unbiased estimations of $V_r^{\pi^i}$ and $V_r^{\pi^{\mathrm{E}}}$, respectively.

$$\widehat{V}_r^{\pi^i} = \sum_{h=1}^{H} r_h(s_h^i, a_h^i), \quad \widehat{V}_r^{\pi^{\mathrm{E}}} = \frac{1}{N}\sum_{\tau \in \mathcal{D}^{\mathrm{E}}}\sum_{h=1}^{H} r_h(\tau(s_h), \tau(a_h)).$$

The first term in the RHS of Eq. (4) is the estimated reward error while the remaining terms are estimation errors. To upper bound the first term, we have

$$
\frac{1}{K}\sum_{k=1}^{K} \widehat{V}_{r^{\mathrm{true}}}^{\pi^{\mathrm{E}}} - \widehat{V}_{r^{\mathrm{true}}}^{\pi^k} - \left(\widehat{V}_{r^k}^{\pi^{\mathrm{E}}} - \widehat{V}_{r^k}^{\pi^k}\right)
$$

$$
= \frac{1}{K}\sum_{k=1}^{K} \widehat{V}_{r^k}^{\pi^k} - \widehat{V}_{r^k}^{\pi^{\mathrm{E}}} - \left(\widehat{V}_{r^{\mathrm{true}}}^{\pi^k} - \widehat{V}_{r^{\mathrm{true}}}^{\pi^{\mathrm{E}}}\right)
$$

$$
\leq \frac{1}{K}\max_{r \in \mathcal{R}}\sum_{k=1}^{K} \widehat{V}_{r^k}^{\pi^{\mathrm{E}}} - \widehat{V}_{r^k}^{\pi^k} - \left(\widehat{V}_r^{\pi^k} - \widehat{V}_r^{\pi^{\mathrm{E}}}\right)
$$

$$
\overset{(c)}{=} \varepsilon_{\mathrm{opt}}^r.
$$

Equation (c) follows the definition of reward optimization error in Definition 2. Then we can obtain

$$
\frac{1}{K}\sum_{k=1}^{K} V_{r^{\mathrm{true}}}^{\pi^{\mathrm{E}}} - V_{r^{\mathrm{true}}}^{\pi^k} - \left(V_{r^k}^{\pi^{\mathrm{E}}} - V_{r^k}^{\pi^k}\right)
$$

$$
\leq V_{r^{\mathrm{true}}}^{\pi^{\mathrm{E}}} - \widehat{V}_{r^{\mathrm{true}}}^{\pi^{\mathrm{E}}} + \frac{1}{K}\sum_{k=1}^{K}\widehat{V}_{r^k}^{\pi^{\mathrm{E}}} - V_{r^k}^{\pi^{\mathrm{E}}} + \frac{1}{K}\sum_{k=1}^{K}\widehat{V}_{r^{\mathrm{true}}}^{\pi^k} - V_{r^{\mathrm{true}}}^{\pi^k} + \frac{1}{K}\sum_{k=1}^{K}V_{r^k}^{\pi^k} - \widehat{V}_{r^k}^{\pi^k} + \varepsilon_{\mathrm{opt}}^r.
$$

Then we proceed to upper bound the estimation errors. First, we first upper bound the estimation error caused by using $\widehat{V}_r^{\pi^{\mathrm{E}}}$ to approximate $V_r^{\pi^{\mathrm{E}}}$. In particular, we have that

$$\left|\widehat{V}_r^{\pi^{\mathrm{E}}} - V_r^{\pi^{\mathrm{E}}}\right| = \left|\frac{1}{N}\sum_{\tau\in\mathcal{D}^{\mathrm{E}}}\sum_{h=1}^{H} r_h(s_h(\tau), a_h(\tau)) - \mathbb{E}\left[\sum_{h=1}^{H} r_h(s_h, a_h)\middle|\pi^{\mathrm{E}}\right]\right|$$

$$= \left|\sum_{h=1}^{H}\frac{1}{N}\sum_{\tau\in\mathcal{D}^{\mathrm{E}}} r_h(s_h(\tau), a_h(\tau)) - \sum_{h=1}^{H}\mathbb{E}\left[r_h(s_h, a_h)\middle|\pi^{\mathrm{E}}\right]\right|$$

$$\leq \sum_{h=1}^{H}\left|\frac{1}{N}\sum_{\tau\in\mathcal{D}^{\mathrm{E}}} r_h(s_h(\tau), a_h(\tau)) - \mathbb{E}\left[r_h(s_h, a_h)\middle|\pi^{\mathrm{E}}\right]\right|.$$

By Hoeffding's inequality [58], for any fixed timestep $h \in [H]$ and any fixed reward function $r_h \in \mathcal{R}_h$, with probability at least $1-\delta$, we have that

$$\left|\frac{1}{N}\sum_{\tau\in\mathcal{D}^{\mathrm{E}}} r_h(s_h(\tau), a_h(\tau)) - \mathbb{E}\left[r_h(s_h, a_h)\middle|\pi^{\mathrm{E}}\right]\right| \leq \sqrt{\frac{\log(2/\delta)}{N}}.$$

Let $(\mathcal{R}_h)_\rho$ be a $\rho$-cover of $\mathcal{R}$. By union bound, with probability at least $1-\delta$, for all $h \in [H]$ and all $\widehat{r}_h \in (\mathcal{R}_h)_\rho$, we have that

$$\left|\frac{1}{N}\sum_{\tau\in\mathcal{D}^{\mathrm{E}}} \widehat{r}_h(s_h(\tau), a_h(\tau)) - \mathbb{E}\left[\widehat{r}_h(s_h, a_h)\middle|\pi^{\mathrm{E}}\right]\right| \leq \sqrt{\frac{\log(2H|(\mathcal{R}_h)_\rho|/\delta)}{N}}.$$

Then with probability at least $1-\delta$, for all $\widehat{r} = (\widehat{r}_1, \ldots, \widehat{r}_H) \in (\mathcal{R}_1)_\rho \times \ldots \times (\mathcal{R}_1)_\rho$,

$$\left|\widehat{V}_{\widehat{r}}^{\pi^{\mathrm{E}}} - V_{\widehat{r}}^{\pi^{\mathrm{E}}}\right| \leq \sum_{h=1}^{H}\left|\frac{1}{N}\sum_{\tau\in\mathcal{D}^{\mathrm{E}}} \widehat{r}_h(s_h(\tau), a_h(\tau)) - \mathbb{E}\left[\widehat{r}_h(s_h, a_h)\middle|\pi^{\mathrm{E}}\right]\right|$$

$$\leq \sum_{h=1}^{H}\sqrt{\frac{\log(2|(\mathcal{R}_h)_\rho|/\delta)}{N}}$$

$$\leq H\sqrt{\frac{\log(2\max_{h\in[H]}|(\mathcal{R}_h)_\rho|/\delta)}{N}}.$$

According to the definition of $\rho$-cover, for any reward function $r = (r_1, \ldots, r_H) \in \mathcal{R}$, there exists $\widehat{r} = (\widehat{r}_1, \ldots, \widehat{r}_H) \in (\mathcal{R}_1)_\rho \times \ldots \times (\mathcal{R}_1)_\rho$ such that $\forall h \in [H]$, $\max_{(s,a)\in\mathcal{S}\times\mathcal{A}}|r_h(s,a) - \widehat{r}_h(s,a)| \leq \rho$. Then we have that

$$\left|\widehat{V}_r^{\pi^{\mathrm{E}}} - \widehat{V}_{\widehat{r}}^{\pi^{\mathrm{E}}}\right| \leq \frac{1}{N}\sum_{\tau\in\mathcal{D}^{\mathrm{E}}}\sum_{h=1}^{H}|r_h(s_h(\tau), a_h(\tau)) - \widehat{r}_h(s_h(\tau), a_h(\tau))| \leq H\rho,$$

$$\left|V_r^{\pi^{\mathrm{E}}} - V_{\widehat{r}}^{\pi^{\mathrm{E}}}\right| \leq \mathbb{E}\left[\sum_{h=1}^{H}|r_h(s_h, a_h) - \widehat{r}_h(s_h, a_h)|\middle|\pi^{\mathrm{E}}\right] \leq H\rho.$$

Then, with probability at least $1-\delta$, for all reward function $r \in \mathcal{R}$, we have that

$$\left|\widehat{V}_r^{\pi^{\mathrm{E}}} - V_r^{\pi^{\mathrm{E}}}\right| \leq \left|\widehat{V}_{\widehat{r}}^{\pi^{\mathrm{E}}} - V_{\widehat{r}}^{\pi^{\mathrm{E}}}\right| + 2H\rho$$

$$\leq H\sqrt{\frac{\log(2\max_{h\in[H]}|(\mathcal{R}_h)_\rho|/\delta)}{N}} + 2H\rho$$

$$\leq H\sqrt{\frac{\log(2\max_{h\in[H]}\mathcal{N}_\rho(\mathcal{R}_h)/\delta)}{N}} + 2H\rho. \tag{5}$$

Now we have obtained the upper bound on the estimation error $|\widehat{V}_r^{\pi^{\mathrm{E}}} - V_r^{\pi^{\mathrm{E}}}|$. Then we proceed to upper bound the estimation error $(1/K)\cdot\sum_{k=1}^{K}\widehat{V}_{r^{\mathrm{true}}}^{\pi^k} - V_{r^{\mathrm{true}}}^{\pi^k}$ and $(1/K)\cdot\sum_{k=1}^{K}V_r^{\pi^k} - \widehat{V}_r^{\pi^k}$. With the Hoeffding's inequality [58, 25], with probability at least $1-\delta$, we obtain that

$$\frac{1}{K}\sum_{k=1}^{K}\widehat{V}_{r^{\mathrm{true}}}^{\pi^k} - V_{r^{\mathrm{true}}}^{\pi^k} \leq H\sqrt{\frac{\log(1/\delta)}{K}}. \tag{6}$$

We proceed to analyze the term $\sum_{k=1}^{K} V_{r^k}^{\pi^k} - \widehat{V}_{r^k}^{\pi^k}$. Notice that $r^k$ are learned from historical trajectories $\{\tau^1, \ldots, \tau^{k-1}\}$ and thus statistically depends on $\{\tau^1, \ldots, \tau^{k-1}\}$. Therefore, $\widehat{V}_{r^1}^{\pi^1}, \cdots, \widehat{V}_{r^k}^{\pi^k}$ are not independent and the standard Hoeffding's inequality is not applicable. To address this issue, we apply Azuma-Hoeffding's inequality [58] for martingale. In particular, we define $\mathcal{F}^k$ as the filtration induced by $\{\tau^1, \cdots, \tau^k\}$ and can obtain that

$$\mathbb{E}\left[V_{r^k}^{\pi^k} - \widehat{V}_{r^k}^{\pi^k} | \mathcal{F}^{k-1}\right] = 0.$$

Therefore, $\{(V_{r^k}^{\pi^k} - \widehat{V}_{r^k}^{\pi^k}, \mathcal{F}^k)\}_{k=1}^{\infty}$ is a martingale difference sequence. With Azuma-Hoeffding's inequality, we can derive that with probability at least $1 - \delta$,

$$\frac{1}{K}\sum_{k=1}^{K} V_{r^k}^{\pi^k} - \widehat{V}_{r^k}^{\pi^k} \leq H\sqrt{\frac{\log(1/\delta)}{K}}. \tag{7}$$

In summary, we have derived the following three high-probability inequalities: Eq. (5), Eq. (6) and Eq. (7). With union bound, with probability at least $1 - \delta$, it holds that

$$\forall r \in \mathcal{R}, \left|\widehat{V}_r^{\pi^E} - V_r^{\pi^E}\right| \leq H\sqrt{\frac{\log(6\max_{h \in [H]}\mathcal{N}_\rho(\mathcal{R}_h)/\delta)}{N}} + 2H\rho,$$

$$\frac{1}{K}\sum_{k=1}^{K}\widehat{V}_{r^{\text{true}}}^{\pi^k} - V_{r^{\text{true}}}^{\pi^k} \leq H\sqrt{\frac{\log(3/\delta)}{K}}, \quad \frac{1}{K}\sum_{k=1}^{K} V_{r^k}^{\pi^k} - \widehat{V}_{r^k}^{\pi^k} \leq H\sqrt{\frac{\log(3/\delta)}{K}}.$$

With the above three inequalities, we can derive that

$$\frac{1}{K}\sum_{k=1}^{K} V_{r^{\text{true}}}^{\pi^E} - V_{r^{\text{true}}}^{\pi^k} - \left(V_{r^k}^{\pi^E} - V_{r^k}^{\pi^k}\right)$$

$$\leq V_{r^{\text{true}}}^{\pi^E} - \widehat{V}_{r^{\text{true}}}^{\pi^E} + \frac{1}{K}\sum_{k=1}^{K}\widehat{V}_{r^k}^{\pi^E} - V_{r^k}^{\pi^E} + \frac{1}{K}\sum_{k=1}^{K}\widehat{V}_{r^{\text{true}}}^{\pi^k} - V_{r^{\text{true}}}^{\pi^k} + \frac{1}{K}\sum_{k=1}^{K} V_{r^k}^{\pi^k} - \widehat{V}_{r^k}^{\pi^k} + \varepsilon_{\text{opt}}^r$$

$$\leq 2H\sqrt{\frac{\log(6\max_{h \in [H]}\mathcal{N}_\rho(\mathcal{R}_h)/\delta)}{N}} + 4H\rho + 2H\sqrt{\frac{\log(3/\delta)}{K}} + \varepsilon_{\text{opt}}^r.$$

We complete the proof.

### B.4 Proof of Lemma 3

To prove Lemma 3, we need the following two auxiliary lemmas. The detailed proof is presented in Appendix B.5 and Appendix B.6.

**Lemma 4.** *For any fixed $\delta \in (0, 1]$, with probability at least $1 - \delta$,*

$$\forall k \in [K], \mathrm{BE}^k(Q^{\star, r^k}) \leq 16H^4 \log\left(KH\max_{h \in [H]}\mathcal{N}_\rho(\mathcal{Q}_h)\mathcal{N}_\rho(\mathcal{R}_h)/\delta\right) + 30kH^3\rho.$$

**Lemma 5.** *For any fixed $\delta \in (0, 1]$, with probability at least $1 - \delta$,*

$$\forall k \in [K], \mathrm{BE}^k(Q^k) \geq \frac{1}{2}\sum_{i=0}^{k-1}\mathbb{E}\left[\sum_{h=1}^{H}\left(Q_h^k(s_h^i, a_h^i) - (\mathcal{T}_h^{r^k}Q_{h+1}^k)(s_h^i, a_h^i)\right)^2 \Big| \pi^i\right]$$

$$- 41H^4 \log\left(2KH\max_{h \in [H]}\mathcal{N}_\rho(\mathcal{Q}_h)\mathcal{N}_\rho(\mathcal{R}_h)/\delta\right) - 27kH^2\rho.$$

Now we proceed to analyze the policy error. First of all, we perform the following error decomposition.

$$\frac{1}{K}\sum_{k=1}^{K} V_{r^k}^{\pi^E} - V_{r^k}^{\pi^k}$$

$$\leq \frac{1}{K} \sum_{k=1}^{K} V_{r^k}^{\star} - V_{r^k}^{\pi^k}$$

$$= \frac{1}{K} \sum_{k=1}^{K} \left( V_{r^k}^{\star} - Q_1^k(s_1, \pi^k) \right) + \frac{1}{K} \sum_{k=1}^{K} \left( Q_1^k(s_1, \pi^k) - V_{r^k}^{\pi^k} \right)$$

$$= \frac{1}{K} \sum_{k=1}^{K} \left( \max_{a \in \mathcal{A}} Q_1^{\star,r^k}(s_1, a) - \max_{a \in \mathcal{A}} Q_1^k(s_1, a) \right) + \frac{1}{K} \sum_{k=1}^{K} \left( Q_1^k(s_1, \pi^k) - V_{r^k}^{\pi^k} \right).$$

Here $V_{r^k}^{\star}$ denotes the optimal policy value under reward $r^k$.

From line 4 in Algorithm 1, we know that $Q^k$ is an approximate solution of $\min_{Q \in \mathcal{Q}} \mathcal{L}^k(Q)$ with an error $\varepsilon_{\text{opt}}^Q$. With $Q^{\star,r^k} \in \mathcal{Q}$ from Assumption 2, we have that

$$\text{BE}^k(Q^k) - \lambda \max_{a \in \mathcal{A}} Q_1^k(s_1, a) \leq \text{BE}^k(Q^{\star,r^k}) - \lambda \max_{a \in \mathcal{A}} Q_1^{\star,r^k}(s_1, a) + \varepsilon_{\text{opt}}^Q.$$

Rearrange the above inequality yields that

$$\max_{a \in \mathcal{A}} Q_1^{\star,r^k}(s_1, a) - \max_{a \in \mathcal{A}} Q_1^k(s_1, a) \leq \frac{1}{\lambda} \left( \text{BE}^k(Q^{\star,r^k}) - \text{BE}^k(Q^k) \right) + \frac{\varepsilon_{\text{opt}}^Q}{\lambda}.$$

From Lemma 4, with probability at least $1 - \delta$, we have

$$\text{BE}^k(Q^{\star,r^k}) \leq 16 H^4 \log \left( KH \max_{h \in [H]} \mathcal{N}_\rho(\mathcal{Q}_h) \mathcal{N}_\rho(\mathcal{R}_h)/\delta \right) + 30 k H^3 \rho.$$

On the other hand, with probability at least $1 - \delta$, we have

$$\text{BE}^k(Q^k) \geq \frac{1}{2} \sum_{i=0}^{k-1} \mathbb{E} \left[ \sum_{h=1}^{H} \left( Q_h^k(s_h^i, a_h^i) - (\mathcal{T}_h^{r^k} Q_{h+1}^k)(s_h^i, a_h^i) \right)^2 \Big| \pi^i \right]$$

$$- 41 H^4 \log \left( 2 KH \max_{h \in [H]} \mathcal{N}_\rho(\mathcal{Q}_h) \mathcal{N}_\rho(\mathcal{R}_h)/\delta \right) - 27 k H^2 \rho.$$

By union bound, with probability at least $1 - \delta$,

$$\max_{a \in \mathcal{A}} Q_1^{\star,r^k}(s_1, a) - \max_{a \in \mathcal{A}} Q_1^k(s_1, a)$$

$$\leq -\frac{1}{2\lambda} \sum_{i=0}^{k-1} \mathbb{E} \left[ \sum_{h=1}^{H} \left( Q_h^k(s_h^i, a_h^i) - (\mathcal{T}_h^{r^k} Q_{h+1}^k)(s_h^i, a_h^i) \right)^2 \Big| \pi^i \right]$$

$$+ \frac{57 H^4 \log(4 KH \max_{h \in [H]} \mathcal{N}_\rho(\mathcal{Q}_h) \mathcal{N}_\rho(\mathcal{R}_h)/\delta) + 57 k H^3 \rho + \varepsilon_{\text{opt}}^Q}{\lambda}.$$

Then we have that

$$\frac{1}{K} \sum_{k=1}^{K} V_{r^k}^{\pi^{\text{E}}} - V_{r^k}^{\pi^k}$$

$$\leq -\frac{1}{2\lambda} \frac{1}{K} \sum_{k=1}^{K} \sum_{i=0}^{k-1} \mathbb{E} \left[ \sum_{h=1}^{H} \left( Q_h^k(s_h^i, a_h^i) - (\mathcal{T}_h^{r^k} Q_{h+1}^k)(s_h^i, a_h^i) \right)^2 \Big| \pi^i \right]$$

$$+ \frac{57 H^4 \log(4 KH \max_{h \in [H]} \mathcal{N}_\rho(\mathcal{Q}_h) \mathcal{N}_\rho(\mathcal{R}_h)/\delta) + 57 K H^3 \rho + \varepsilon_{\text{opt}}^Q}{\lambda}$$

$$+ \frac{1}{K} \sum_{k=1}^{K} \left( Q_1^k(s_1, \pi^k) - V_{r^k}^{\pi^k} \right).$$

Now we upper bound the last term in RHS of the above inequality. From Assumption 4, for any $\mu \geq 0$, it holds that

$$\frac{1}{K} \sum_{k=1}^{K} Q_1^k(s_1, \pi^k) - V_{r^k}^{\pi^k} \leq \frac{\mu}{2K} \sum_{k=1}^{K} \sum_{i=1}^{k-1} \mathbb{E} \left[ \sum_{h=1}^{H} \left( Q_h^k(s_h, a_h) - \mathcal{T}_h^{r^k} Q_{h+1}^k(s_h, a_h) \right)^2 \Big| \pi^i \right] + \frac{d}{2\mu K}$$

$$+ \sqrt{\frac{dH}{K}} + \varepsilon H$$

$$= \frac{1}{2\lambda K} \sum_{k=1}^{K} \sum_{i=1}^{k-1} \mathbb{E} \left[ \sum_{h=1}^{H} \left( Q_h^k(s_h, a_h) - \mathcal{T}_h^{r^k} Q_{h+1}^k(s_h, a_h) \right)^2 \bigg| \pi^i \right] + \frac{\lambda d}{2K}$$

$$+ \sqrt{\frac{dH}{K}} + \varepsilon H.$$

The last equation is obtained by setting $\mu = 1/\lambda$. Combining the above two inequalities yields that

$$\frac{1}{K} \sum_{k=1}^{K} V_{r^k}^{\pi^E} - V_{r^k}^{\pi^k}$$

$$\leq \frac{57H^4 \log(4KH \max_{h \in [H]} \mathcal{N}_\rho(\mathcal{Q}_h) \mathcal{N}_\rho(\mathcal{R}_h)/\delta) + 57KH^3 \rho + \varepsilon_{\text{opt}}^Q}{\lambda} + \frac{\lambda d}{2K} + \sqrt{\frac{dH}{K}} + \varepsilon H.$$

## B.5 Proof of Lemma 4

Recall the definition of the estimated Bellman error.

$$\text{BE}^k(Q^{\star, r^k}) = \sum_{h=1}^{H} \mathcal{E}_h(Q_h^{\star, r^k}, Q_{h+1}^{\star, r^k}; \mathcal{D}^k, r^k) - \inf_{Q_h' \in \mathcal{Q}_h} \text{BE}_h(Q_h', Q_{h+1}^{\star, r^k}; \mathcal{D}^k, r^k)$$

$$= \sum_{h=1}^{H} \sum_{i=0}^{k-1} \left( Q_h^{\star, r^k}(s_h^i, a_h^i) - r_h^k(s_h^i, a_h^i) - \max_{a'} Q_{h+1}^{\star, r^k}(s_{h+1}^i, a') \right)^2$$

$$- \inf_{Q_h' \in \mathcal{Q}_h} \sum_{i=0}^{k-1} \left( Q_h'(s_h^i, a_h^i) - r_h^k(s_h^i, a_h^i) - \max_{a'} Q_{h+1}^{\star, r^k}(s_{h+1}^i, a') \right)^2.$$

For any fixed tuple $(k, h, Q', r) \in [K] \times [H] \times \mathcal{Q} \times \mathcal{R}$, we define the random variable

$$Z_h^i(Q', r) := \left( Q_h'(s_h^i, a_h^i) - r_h(s_h^i, a_h^i) - \max_{a' \in \mathcal{A}} Q_{h+1}^{\star, r}(s_{h+1}^i, a') \right)^2$$

$$- \left( Q_h^{\star, r}(s_h^i, a_h^i) - r_h(s_h^i, a_h^i) - \max_{a' \in \mathcal{A}} Q_{h+1}^{\star, r}(s_{h+1}^i, a') \right)^2.$$

Furthermore, we define the filtration $\mathcal{F}_h^i = \sigma(\{(s_1^j, a_1^j, \ldots, s_H^j, a_H^j)\}_{j=0}^{i-1} \cup \{s_1^i, a_1^i, \ldots, s_h^i, a_h^i\})$.
Then we calculate the expectation and variance of $Z_h^i(Q', r)$ conditioned on $\mathcal{F}_h^i$.

$$\mathbb{E}\left[ Z_h^i(Q', r) | \mathcal{F}_h^i \right]$$

$$= \mathbb{E}\left[ \left( Q_h'(s_h^i, a_h^i) - Q_h^{\star, r}(s_h^i, a_h^i) + Q_h^{\star, r}(s_h^i, a_h^i) - r_h(s_h^i, a_h^i) - \max_{a' \in \mathcal{A}} Q_{h+1}^{\star, r}(s_{h+1}^i, a') \right)^2 \bigg| \mathcal{F}_h^i \right]$$

$$- \mathbb{E}\left[ \left( Q_h^{\star, r}(s_h^i, a_h^i) - r_h(s_h^i, a_h^i) - \max_{a' \in \mathcal{A}} Q_{h+1}^{\star, r}(s_{h+1}^i, a') \right)^2 \bigg| \mathcal{F}_h^i \right]$$

$$= \mathbb{E}\left[ \left( Q_h'(s_h^i, a_h^i) - Q_h^{\star, r}(s_h^i, a_h^i) \right)^2 \bigg| \mathcal{F}_h^i \right]$$

$$+ 2\mathbb{E}\left[ \left( Q_h'(s_h^i, a_h^i) - Q_h^{\star, r}(s_h^i, a_h^i) \right) \left( Q_h^{\star, r}(s_h^i, a_h^i) - r_h(s_h^i, a_h^i) - \max_{a' \in \mathcal{A}} Q_{h+1}^{\star, r}(s_{h+1}^i, a') \right) \bigg| \mathcal{F}_h^i \right]$$

$$= \left( Q_h'(s_h^i, a_h^i) - Q_h^{\star, r}(s_h^i, a_h^i) \right)^2$$

$$+ 2 \left( Q_h'(s_h^i, a_h^i) - Q_h^{\star, r}(s_h^i, a_h^i) \right) \mathbb{E}\left[ Q_h^{\star, r}(s_h^i, a_h^i) - r_h(s_h^i, a_h^i) - \max_{a' \in \mathcal{A}} Q_{h+1}^{\star, r}(s_{h+1}^i, a') \bigg| \mathcal{F}_h^i \right]$$

$$= \left( Q_h'(s_h^i, a_h^i) - Q_h^{\star, r}(s_h^i, a_h^i) \right)^2$$

$$+ 2\left(Q'_h(s^i_h, a^i_h) - Q^{\star,r}_h(s^i_h, a^i_h)\right)\left(Q^{\star,r}_h(s^i_h, a^i_h) - (\mathcal{T}^r_h Q^{\star,r}_{h+1})(s^i_h, a^i_h)\right)$$
$$= \left(Q'_h(s^i_h, a^i_h) - Q^{\star,r}_h(s^i_h, a^i_h)\right)^2.$$

For the conditional variance, we have that

$$\mathrm{Var}\left[Z^i_h(Q', r)\Big|\mathcal{F}^i_h\right]$$

$$\leq \mathbb{E}\left[\left(Z^i_h(Q', r)\right)^2\Big|\mathcal{F}^i_h\right]$$

$$= \mathbb{E}\Bigg[\left(Q'_h(s^i_h, a^i_h) - Q^{\star,r}_h(s^i_h, a^i_h)\right)^2 \cdot$$
$$\left(Q'_h(s^i_h, a^i_h) + Q^{\star,r}_h(s^i_h, a^i_h) - 2\left(r_h(s^i_h, a^i_h) + \max_{a'\in\mathcal{A}} Q^{\star,r}_{h+1}(s^i_{h+1}, a')\right)\right)^2\Big|\mathcal{F}^i_h\Bigg]$$

$$\overset{(a)}{\leq} 16H^2\left(Q'_h(s^i_h, a^i_h) - Q^{\star,r}_h(s^i_h, a^i_h)\right)^2$$

$$= 16H^2\mathbb{E}\left[Z^i_h(Q', r)\Big|\mathcal{F}^i_h\right].$$

Here inequality $(a)$ holds since $|Q'_h(s^i_h, a^i_h) + Q^{\star,r}_h(s^i_h, a^i_h) - 2(r_h(s^i_h, a^i_h) + \max_{a'\in\mathcal{A}} Q^{\star,r}_{h+1}(s^i_{h+1}, a'))| \leq 4H$ almost surely.

Notice that $\{Z^i_h(Q', r) - \mathbb{E}\left[Z^i_h(Q', r)|\mathcal{F}^i_h\right]\}^{k-1}_{i=0}$ is the martingale difference sequence adapted to $\{\mathcal{F}^i_h\}^{k-1}_{i=0}$. Besides, almost surely, we have that

$$\left|Z^i_h(Q', r)\right| \leq \max\Bigg\{\left(Q'_h(s^i_h, a^i_h) - r_h(s^i_h, a^i_h) - \max_{a'\in\mathcal{A}} Q^{\star,r}_{h+1}(s^i_{h+1}, a')\right)^2,$$
$$\left(Q^{\star,r}_h(s^i_h, a^i_h) - r_h(s^i_h, a^i_h) - \max_{a'\in\mathcal{A}} Q^{\star,r}_{h+1}(s^i_{h+1}, a')\right)^2\Bigg\}$$
$$\leq 4H^2.$$

Then we immediately get that $|Z^i_h(Q', r) - \mathbb{E}\left[Z^i_h(Q', r)|\mathcal{F}^i_h\right]| \leq 8H^2$ almost surely. Thus we can apply Lemma 6 and obtain that for any $\eta \in (0, 1/(4H^2)]$, with probability at least $1 - \delta$,

$$\left|\sum^{k-1}_{i=0} Z^i_h(Q', r) - \sum^{k-1}_{i=0} \mathbb{E}\left[Z^i_h(Q', r)\Big|\mathcal{F}^i_h\right]\right|$$

$$\leq \eta\sum^{k-1}_{i=0}\mathrm{Var}\left[Z^i_h(Q', r)\Big|\mathcal{F}^i_h\right] + \frac{\log(1/\delta)}{\eta}$$

$$\leq 36H^2\eta\sum^{k-1}_{i=0}\mathbb{E}\left[Z^i_h(Q', r)\Big|\mathcal{F}^i_h\right] + \frac{\log(1/\delta)}{\eta}.$$

This implies that

$$-\sum^{k-1}_{i=0} Z^i_h(Q', r) \leq \left(36H^2\eta - 1\right)\sum^{k-1}_{i=0}\mathbb{E}\left[Z^i_h(Q', r)\Big|\mathcal{F}^i_h\right] + \frac{\log(1/\delta)}{\eta}$$

$$\leq 16H^2\log(1/\delta).$$

The last equation is obtained by choosing $\eta = 1/(16H^2)$.

We define $(\mathcal{Q}_h)_\rho$ and $(\mathcal{R}_h)_\rho$ as the $\rho$-cover of $\mathcal{Q}_h$ and $\mathcal{R}_h$, respectively. It is direct to have that $\mathcal{Q}_\rho = (\mathcal{Q}_1)_\rho \times \ldots (\mathcal{Q}_H)_\rho$ and $\mathcal{R}_\rho = (\mathcal{R}_1)_\rho \times \ldots (\mathcal{R}_H)_\rho$ are $\rho$-covers of $\mathcal{Q}$ and $\mathcal{R}$, respectively. By union bound, with probability at least $1 - \delta$, for all $(k, h, \widehat{Q}, \widehat{r}) \in [K] \times [H] \times \mathcal{Q}_\rho \times \mathcal{R}_\rho$, we have that

$$-\sum^{k-1}_{i=0} Z^i_h(\widehat{Q}, \widehat{r}) \leq 16H^2\log\left(KH\prod^H_{h=1}(|(\mathcal{Q}_h)_\rho||(\mathcal{R}_h)_\rho|)/\delta\right)$$

$$\leq 16H^3 \log\left(KH \max_{h\in[H]} |(\mathcal{Q}_h)_\rho||(\mathcal{R}_h)_\rho|/\delta\right).$$

Furthermore, for any $(Q,r) \in \mathcal{Q} \times \mathcal{R}$, there exists $(\widehat{Q},\widehat{r}) \in \mathcal{Q}_\rho \times \mathcal{R}_\rho$ such that $\|Q - \widehat{Q}\|_\infty \leq \rho$ and $\|r - \widehat{r}\|_\infty \leq \rho$. Then we have that

$$\left|\sum_{i=0}^{k-1} Z_h^i(Q,r) - \sum_{i=0}^{k-1} Z_h^i(\widehat{Q},\widehat{r})\right| \leq \sum_{i=0}^{k-1} \left|Z_h^i(Q,r) - Z_h^i(\widehat{Q},\widehat{r})\right|.$$

For each term, we have that

$$\left|Z_h^i(Q,r) - Z_h^i(\widehat{Q},\widehat{r})\right|$$

$$\leq \left|\left(Q_h(s_h^i,a_h^i) - r_h(s_h^i,a_h^i) - \max_{a'\in\mathcal{A}} Q_{h+1}^{\star,r}(s_{h+1}^i,a')\right)^2\right.$$

$$\left. - \left(\widehat{Q}_h(s_h^i,a_h^i) - \widehat{r}_h(s_h^i,a_h^i) - \max_{a'\in\mathcal{A}} Q_{h+1}^{\star,\widehat{r}}(s_{h+1}^i,a')\right)^2\right|$$

$$+ \left|\left(Q_h^{\star,r}(s_h^i,a_h^i) - r_h(s_h^i,a_h^i) - \max_{a'\in\mathcal{A}} Q_{h+1}^{\star,r}(s_{h+1}^i,a')\right)^2\right.$$

$$\left. - \left(Q_h^{\star,\widehat{r}}(s_h^i,a_h^i) - \widehat{r}_h(s_h^i,a_h^i) - \max_{a'\in\mathcal{A}} Q_{h+1}^{\star,\widehat{r}}(s_{h+1}^i,a')\right)^2\right|.$$

For the first term in RHS, we have that

$$\left|\left(Q_h(s_h^i,a_h^i) - r_h(s_h^i,a_h^i) - \max_{a'\in\mathcal{A}} Q_{h+1}^{\star,r}(s_{h+1}^i,a')\right)^2\right.$$

$$\left. - \left(\widehat{Q}_h(s_h^i,a_h^i) - \widehat{r}_h(s_h^i,a_h^i) - \max_{a'\in\mathcal{A}} Q_{h+1}^{\star,\widehat{r}}(s_{h+1}^i,a')\right)^2\right|$$

$$\leq \left|Q_h(s_h^i,a_h^i) - r_h(s_h^i,a_h^i) - \max_{a'\in\mathcal{A}} Q_{h+1}^{\star,r}(s_{h+1}^i,a') + \widehat{Q}_h(s_h^i,a_h^i) - \widehat{r}_h(s_h^i,a_h^i) - \max_{a'\in\mathcal{A}} Q_{h+1}^{\star,\widehat{r}}(s_{h+1}^i,a')\right|$$

$$\left|Q_h(s_h^i,a_h^i) - \widehat{Q}_h(s_h^i,a_h^i) - r_h(s_h^i,a_h^i) + \widehat{r}_h(s_h^i,a_h^i) - \max_{a'\in\mathcal{A}} Q_{h+1}^{\star,r}(s_{h+1}^i,a') + \max_{a'\in\mathcal{A}} Q_{h+1}^{\star,\widehat{r}}(s_{h+1}^i,a')\right|$$

$$\leq 4H\left(\left|Q_h(s_h^i,a_h^i) - \widehat{Q}_h(s_h^i,a_h^i)\right| + \left|r_h(s_h^i,a_h^i) - \widehat{r}_h(s_h^i,a_h^i)\right|\right.$$

$$\left. + \max_{a'\in\mathcal{A}} \left|Q_{h+1}^{\star,r}(s_{h+1}^i,a') - Q_{h+1}^{\star,\widehat{r}}(s_{h+1}^i,a')\right|\right)$$

$$\leq 12H^2\rho.$$

The last inequality follows Lemma 7. Similarly, for the second term in RHS, we have that

$$\left|\left(Q_h^{\star,r}(s_h^i,a_h^i) - r_h(s_h^i,a_h^i) - \max_{a'\in\mathcal{A}} Q_{h+1}^{\star,r}(s_{h+1}^i,a')\right)^2\right.$$

$$\left. - \left(Q_h^{\star,\widehat{r}}(s_h^i,a_h^i) - \widehat{r}_h(s_h^i,a_h^i) - \max_{a'\in\mathcal{A}} Q_{h+1}^{\star,\widehat{r}}(s_{h+1}^i,a')\right)^2\right|$$

$$\leq \left|Q_h^{\star,r}(s_h^i,a_h^i) - r_h(s_h^i,a_h^i) - \max_{a'\in\mathcal{A}} Q_{h+1}^{\star,r}(s_{h+1}^i,a')\right.$$

$$\left. + Q_h^{\star,\widehat{r}}(s_h^i,a_h^i) - \widehat{r}_h(s_h^i,a_h^i) - \max_{a'\in\mathcal{A}} Q_{h+1}^{\star,\widehat{r}}(s_{h+1}^i,a')\right|$$

$$\cdot \left|Q_h^{\star,r}(s_h^i,a_h^i) - Q_h^{\star,\widehat{r}}(s_h^i,a_h^i) - r_h(s_h^i,a_h^i) + \widehat{r}_h(s_h^i,a_h^i)\right.$$

$$\left. - \max_{a'\in\mathcal{A}} Q_{h+1}^{\star,r}(s_{h+1}^i,a') + \max_{a'\in\mathcal{A}} Q_{h+1}^{\star,\widehat{r}}(s_{h+1}^i,a')\right|$$

$$\leq 6H\left(\left|Q_h^{\star,r}(s_h^i, a_h^i) - Q_h^{\star,\widehat{r}}(s_h^i, a_h^i)\right| + \left|r_h(s_h^i, a_h^i) - \widehat{r}_h(s_h^i, a_h^i)\right|\right.$$
$$\left. + \max_{a'\in\mathcal{A}}\left|Q_{h+1}^{\star,r}(s_{h+1}^i, a') - Q_{h+1}^{\star,\widehat{r}}(s_{h+1}^i, a')\right|\right)$$
$$\leq 18H^2\rho.$$

Combining the above four inequalities yields that

$$\left|\sum_{i=0}^{k-1} Z_h^i(Q,r) - \sum_{i=0}^{k-1} Z_h^i(\widehat{Q}, \widehat{r})\right| \leq \sum_{i=0}^{k-1}\left|Z_h^i(Q,r) - Z_h^i(\widehat{Q}, \widehat{r})\right| \leq 30kH^2\rho.$$

Therefore, for all $(Q, r) \in \mathcal{Q} \times \mathcal{R}$,

$$-\sum_{i=0}^{k-1} Z_h^i(Q,r) \leq -\sum_{i=0}^{k-1} Z_h^i(\widehat{Q}, \widehat{r}) + \left|\sum_{i=0}^{k-1} Z_h^i(Q,r) - \sum_{i=0}^{k-1} Z_h^i(\widehat{Q}, \widehat{r})\right|$$
$$\leq 16H^3\log(KH\max_{h\in[H]}|(\mathcal{Q}_h)_\rho||(\mathcal{R}_h)_\rho|/\delta) + 30kH^2\rho$$
$$\leq 16H^3\log(KH\max_{h\in[H]}\mathcal{N}_\rho(\mathcal{Q}_h)\mathcal{N}_\rho(\mathcal{R}_h)/\delta) + 30kH^2\rho.$$

This implies that

$$\left(Q_h^{\star,r}(s_h^i, a_h^i) - r_h(s_h^i, a_h^i) - \max_{a'\in\mathcal{A}}Q_{h+1}^{\star,r}(s_{h+1}^i, a')\right)^2$$
$$\leq \inf_{Q_h\in\mathcal{Q}_h}\left(Q_h(s_h^i, a_h^i) - r_h(s_h^i, a_h^i) - \max_{a'\in\mathcal{A}}Q_{h+1}^{\star,r}(s_{h+1}^i, a')\right)^2$$
$$+ 16H^3\log(KH\max_{h\in[H]}\mathcal{N}_\rho(\mathcal{Q}_h)\mathcal{N}_\rho(\mathcal{R}_h)/\delta) + 30kH^2\rho.$$

Therefore, we can derive the upper bound on $\mathrm{BE}^k(Q^{\star,r^k})$.

$$\mathrm{BE}^k(Q^{\star,r^k})$$
$$= \sum_{h=1}^{H}\left(\sum_{i=0}^{k-1}\left(Q_h^{\star,r^k}(s_h^i, a_h^i) - r_h^k(s_h^i, a_h^i) - \max_{a'}Q_{h+1}^{\star,r^k}(s_{h+1}^i, a')\right)^2\right.$$
$$\left. - \inf_{Q_h'\in\mathcal{Q}_h}\sum_{i=0}^{k-1}\left(Q_h'(s_h^i, a_h^i) - r_h^k(s_h^i, a_h^i) - \max_{a'}Q_{h+1}^{\star,r^k}(s_{h+1}^i, a')\right)^2\right)$$
$$\leq 16H^4\log(KH\max_{h\in[H]}\mathcal{N}_\rho(\mathcal{Q}_h)\mathcal{N}_\rho(\mathcal{R}_h)/\delta) + 30kH^3\rho.$$

We complete the proof.

### B.6 Proof of Lemma 5

For any fixed tuple $(k, h, Q, r) \in [K] \times [H] \times \mathcal{Q} \times \mathcal{R}$, we define the random variable.

$$X_h^i(Q, r) := \left(Q_h(s_h^i, a_h^i) - r_h(s_h^i, a_h^i) - \max_{a'}Q_{h+1}(s_{h+1}^i, a')\right)^2$$
$$- \left((\mathcal{T}_h^r Q_{h+1})(s_h^i, a_h^i) - r_h(s_h^i, a_h^i) - \max_{a'}Q_{h+1}(s_{h+1}^i, a')\right)^2.$$

We define the filtration $\mathcal{F}^i = \sigma(\{(s_1^j, a_1^j, \ldots, s_H^j, a_H^j)\}_{j=0}^{i-1})$. In the following part, we calculate the expectation and variance of $X_h^i(Q, r)$ conditioned on $\mathcal{F}^i$.

$$\mathbb{E}\left[X_h^i(Q, r)|\mathcal{F}^i\right]$$
$$= \mathbb{E}\left[\left(Q_h(s_h^i, a_h^i) - r_h(s_h^i, a_h^i) - \max_{a'}Q_{h+1}(s_{h+1}^i, a')\right)^2\middle|\mathcal{F}^i\right]$$

$$- \mathbb{E}\left[\left((\mathcal{T}_h^r Q_{h+1})(s_h^i, a_h^i) - r_h(s_h^i, a_h^i) - \max_{a'} Q_{h+1}(s_{h+1}^i, a')\right)^2 \Big| \mathcal{F}^i\right]$$

$$= \mathbb{E}\left[\left(Q_h(s_h^i, a_h^i) - (\mathcal{T}_h^r Q_{h+1})(s_h^i, a_h^i) + (\mathcal{T}_h^r Q_{h+1})(s_h^i, a_h^i) - r_h(s_h^i, a_h^i) - \max_{a'} Q_{h+1}(s_{h+1}^i, a')\right)^2 \Big| \mathcal{F}^i\right]$$

$$- \mathbb{E}\left[\left((\mathcal{T}_h^r Q_{h+1})(s_h^i, a_h^i) - r_h(s_h^i, a_h^i) - \max_{a'} Q_{h+1}(s_{h+1}^i, a')\right)^2 \Big| \mathcal{F}^i\right]$$

$$= \mathbb{E}\left[\left(Q_h(s_h^i, a_h^i) - (\mathcal{T}_h^r Q_{h+1})(s_h^i, a_h^i)\right)^2 \Big| \mathcal{F}^i\right]$$

$$+ 2\mathbb{E}\left[\left(Q_h(s_h^i, a_h^i) - (\mathcal{T}_h^r Q_{h+1})(s_h^i, a_h^i)\right)\left((\mathcal{T}_h^r Q_{h+1})(s_h^i, a_h^i) - r_h(s_h^i, a_h^i) - \max_{a'} Q_{h+1}(s_{h+1}^i, a')\right) \Big| \mathcal{F}^i\right]$$

$$= \mathbb{E}\left[\left(Q_h(s_h^i, a_h^i) - (\mathcal{T}_h^r Q_{h+1})(s_h^i, a_h^i)\right)^2 \Big| \mathcal{F}^i\right]$$

$$+ 2\mathbb{E}\bigg[\left(Q_h(s_h^i, a_h^i) - (\mathcal{T}_h^r Q_{h+1})(s_h^i, a_h^i)\right)$$

$$\cdot \mathbb{E}\left[\left((\mathcal{T}_h^r Q_{h+1})(s_h^i, a_h^i) - r_h(s_h^i, a_h^i) - \max_{a'} Q_{h+1}(s_{h+1}^i, a')\right) \Big| s_h^i, a_h^i\right] \Big| \mathcal{F}^i\bigg]$$

$$= \mathbb{E}\left[\left(Q_h(s_h^i, a_h^i) - (\mathcal{T}_h^r Q_{h+1})(s_h^i, a_h^i)\right)^2 \Big| \pi^i\right].$$

$$\mathrm{Var}\left[X_h^i(Q, r) | \mathcal{F}^i\right]$$

$$\leq \mathbb{E}\left[\left(X_h^i(Q, r)\right)^2 | \mathcal{F}^i\right]$$

$$= \mathbb{E}\bigg[\left(Q_h(s_h^i, a_h^i) + (\mathcal{T}_h^r Q_{h+1})(s_h^i, a_h^i) - 2r_h(s_h^i, a_h^i) - 2\max_{a'} Q_{h+1}(s_{h+1}^i, a')\right)^2$$

$$\cdot \left(Q_h(s_h^i, a_h^i) - (\mathcal{T}_h^r Q_{h+1})(s_h^i, a_h^i)\right)^2 \Big| \mathcal{F}^i\bigg]$$

$$\leq 16H^2 \mathbb{E}\left[\left(Q_h(s_h^i, a_h^i) - (\mathcal{T}_h^r Q_{h+1})(s_h^i, a_h^i)\right)^2 \Big| \pi^i\right]$$

$$= 16H^2 \mathbb{E}\left[X_h^i(Q, r) | \mathcal{F}^i\right].$$

Furthermore, $\{X_h^i(Q, r) - \mathbb{E}[X_h^i(Q, r) | \mathcal{F}^i]\}_{i=0}^{k-1}$ is a martingale difference sequence adapted to $\{\mathcal{F}^i\}_{i=0}^{k-1}$. Besides, it is easy to obtain that $|X_h^i(Q, r)| \leq 9H^2$ almost surely. Thus, we can apply Lemma 6 and obtain that with probability at least $1 - \delta$, for any $\eta \in (0, 1/(9H^2)]$,

$$\left|\sum_{i=0}^{k-1} X_h^i(Q, r) - \sum_{i=0}^{k-1} \mathbb{E}[X_h^i(Q, r) | \mathcal{F}^i]\right| \leq \eta \sum_{i=0}^{k-1} \mathrm{Var}\left[X_h^i(Q, r) | \mathcal{F}^i\right] + \frac{\log(2/\delta)}{\eta}$$

$$\leq 16H^2 \eta \sum_{i=0}^{k-1} \mathbb{E}\left[X_h^i(Q, r) | \mathcal{F}^i\right] + \frac{\log(2/\delta)}{\eta}.$$

By choosing $\eta = \min\{1/(9H^2), \sqrt{\log(2/\delta)/(16H^2 \sum_{i=0}^{k-1} \mathbb{E}\left[X_h^i(Q, r) | \mathcal{F}^i\right])}\}$, we have that

$$\left|\sum_{i=0}^{k-1} X_h^i(Q, r) - \sum_{i=0}^{k-1} \mathbb{E}[X_h^i(Q, r) | \mathcal{F}^i]\right| \leq 8H \sqrt{\sum_{i=0}^{k-1} \mathbb{E}\left[X_h^i(Q, r) | \mathcal{F}^i\right] \log(2/\delta)} + 9H^2 \log(2/\delta).$$

This implies that

$$\sum_{i=0}^{k-1} \mathbb{E}[X_h^i(Q, r) | \mathcal{F}^i] - 8H \sqrt{\sum_{i=0}^{k-1} \mathbb{E}\left[X_h^i(Q, r) | \mathcal{F}^i\right] \log(2/\delta)} \leq \sum_{i=0}^{k-1} X_h^i(Q, r) + 9H^2 \log(2/\delta).$$

This establishes a quadratic formula of $x^2 - bx - c \leq 0$ with $x = \sqrt{\sum_{i=0}^{k-1} \mathbb{E}[X_h^i(Q,r)|\mathcal{F}^i]}$, $b = 8H\sqrt{\log(2/\delta)}$ and $c = \sum_{i=0}^{k-1} X_h^i(Q,r) + 9H^2 \log(2/\delta)$. Solving this quadratic formula yields that $(b - \sqrt{b^2 + 4c})/2 \leq x \leq (b + \sqrt{b^2 + 4c})/2$, which implies that

$$x^2 \leq \frac{(b + \sqrt{b^2 + 4c})^2}{4} \leq \frac{2\left(b^2 + b^2 + 4c\right)}{4} = b^2 + 2c.$$

Thus we obtain that

$$\sum_{i=0}^{k-1} \mathbb{E}[X_h^i(Q,r)|\mathcal{F}^i] \leq 2\sum_{i=0}^{k-1} X_h^i(Q,r) + 82H^2 \log(2/\delta).$$

We define $(\mathcal{Q}_h)_\rho$ and $(\mathcal{R}_h)_\rho$ as the $\rho$-covers of $\mathcal{Q}_h$ and $\mathcal{R}_h$, respectively. It is direct to have that $\mathcal{Q}_\rho = (\mathcal{Q}_1)_\rho \times \ldots (\mathcal{Q}_H)_\rho$ and $\mathcal{R}_\rho = (\mathcal{R}_1)_\rho \times \ldots (\mathcal{R}_H)_\rho$ are $\rho$-covers of $\mathcal{Q}$ and $\mathcal{R}$, respectively. By union bound, with probability at least $1 - \delta$, for all $(k, h, \widehat{Q}, \widehat{r}) \in [K] \times [H] \times \mathcal{Q}_\rho \times \mathcal{R}_\rho$,

$$\sum_{i=0}^{k-1} \mathbb{E}[X_h^i(\widehat{Q},\widehat{r})|\mathcal{F}^i] \leq 2\sum_{i=0}^{k-1} X_h^i(\widehat{Q},\widehat{r}) + 82H^2 \log(2KH|\mathcal{Q}_\rho||\mathcal{R}_\rho|/\delta)$$

$$= 2\sum_{i=0}^{k-1} X_h^i(\widehat{Q},\widehat{r}) + 82H^2 \log\left(2KH\prod_{h=1}^{H} \left(|(\mathcal{Q}_h)_\rho||(\mathcal{R}_h)_\rho|\right)/\delta\right)$$

$$\leq 2\sum_{i=0}^{k-1} X_h^i(\widehat{Q},\widehat{r}) + 82H^3 \log\left(2KH\max_{h\in[H]} |(\mathcal{Q}_h)_\rho||(\mathcal{R}_h)_\rho|/\delta\right).$$

We have calculated the conditional expectation in the LHS and obtain that

$$\sum_{i=0}^{k-1} \mathbb{E}\left[\left(\widehat{Q}_h(s_h^i, a_h^i) - (\mathcal{T}_h^{\widehat{r}}\widehat{Q}_{h+1})(s_h^i, a_h^i)\right)^2 \middle| \pi^i\right]$$

$$\leq 2\sum_{i=0}^{k-1} X_h^i(\widehat{Q},\widehat{r}) + 82H^3 \log\left(2KH\max_{h\in[H]} |(\mathcal{Q}_h)_\rho||(\mathcal{R}_h)_\rho|/\delta\right)$$

$$\leq 2\sum_{i=0}^{k-1} X_h^i(\widehat{Q},\widehat{r}) + 82H^3 \log\left(2KH\max_{h\in[H]} \mathcal{N}_\rho(\mathcal{Q}_h)\mathcal{N}_\rho(\mathcal{R}_h)/\delta\right).$$

According to the definition of $\rho$-cover, for $(Q^k, r^k)$, there exists $(\widehat{Q}, \widehat{r}) \in \mathcal{Q}_\rho \times \mathcal{R}_\rho$ such that

$$\max_{(s,a,h)\in\mathcal{S}\times\mathcal{A}\times[H]} \left|\widehat{Q}_h(s,a) - Q_h^k(s,a)\right| \leq \rho, \quad \max_{(s,a,h)\in\mathcal{S}\times\mathcal{A}\times[H]} \left|\widehat{r}_h(s,a) - r_h^k(s,a)\right| \leq \rho.$$

Then we can upper bound the errors caused by approximating $(Q^k, r^k)$ with $(\widehat{Q}, \widehat{r})$.

$$\left|\left(\widehat{Q}_h(s_h^i, a_h^i) - (\mathcal{T}_h^{\widehat{r}}\widehat{Q}_{h+1})(s_h^i, a_h^i)\right)^2 - \left(Q_h^k(s_h^i, a_h^i) - (\mathcal{T}_h^{r^k}Q_{h+1}^k)(s_h^i, a_h^i)\right)^2\right|$$

$$\leq \left|\widehat{Q}_h(s_h^i, a_h^i) - (\mathcal{T}_h^{\widehat{r}}\widehat{Q}_{h+1})(s_h^i, a_h^i) + Q_h^k(s_h^i, a_h^i) - (\mathcal{T}_h^{r^k}Q_{h+1}^k)(s_h^i, a_h^i)\right|$$

$$\left|\widehat{Q}_h(s_h^i, a_h^i) - (\mathcal{T}_h^{\widehat{r}}\widehat{Q}_{h+1})(s_h^i, a_h^i) - Q_h^k(s_h^i, a_h^i) + (\mathcal{T}_h^{r^k}Q_{h+1}^k)(s_h^i, a_h^i)\right|$$

$$\leq 2H \left|\widehat{Q}_h(s_h^i, a_h^i) - (\mathcal{T}_h^{\widehat{r}}\widehat{Q}_{h+1})(s_h^i, a_h^i) - Q_h^k(s_h^i, a_h^i) + (\mathcal{T}_h^{r^k}Q_{h+1}^k)(s_h^i, a_h^i)\right|$$

$$\leq 6H\rho.$$

$$\left|X_h^i(\widehat{Q},\widehat{r}) - X_h^i(Q^k, r^k)\right|$$

$$\leq \left|\widehat{Q}_h(s_h^i, a_h^i) + Q_h^k(s_h^i, a_h^i) - \widehat{r}_h(s_h^i, a_h^i) - r_h^k(s_h^i, a_h^i)\right.$$

$$- \max_{a'} \widehat{Q}_{h+1}(s_{h+1}^i, a') - \max_{a'} Q_{h+1}^k(s_{h+1}^i, a') \Big|$$

$$\cdot \Big| \widehat{Q}_h(s_h^i, a_h^i) - Q_h^k(s_h^i, a_h^i) - \widehat{r}_h(s_h^i, a_h^i) + r_h^k(s_h^i, a_h^i)$$

$$- \max_{a'} \widehat{Q}_{h+1}(s_{h+1}^i, a') + \max_{a'} Q_{h+1}^k(s_{h+1}^i, a') \Big|$$

$$+ \Big| (\mathcal{T}_h^{\widehat{r}} \widehat{Q}_{h+1})(s_h^i, a_h^i) + (\mathcal{T}_h^{r^k} Q_{h+1}^k)(s_h^i, a_h^i) - \widehat{r}_h(s_h^i, a_h^i) - r_h^k(s_h^i, a_h^i)$$

$$- \max_{a'} \widehat{Q}_{h+1}(s_{h+1}^i, a') - \max_{a'} Q_{h+1}^k(s_{h+1}^i, a') \Big|$$

$$\cdot \Big| (\mathcal{T}_h^{\widehat{r}} \widehat{Q}_{h+1})(s_h^i, a_h^i) - (\mathcal{T}_h^{r^k} Q_{h+1}^k)(s_h^i, a_h^i) - \widehat{r}_h(s_h^i, a_h^i) + r_h^k(s_h^i, a_h^i)$$

$$- \max_{a'} \widehat{Q}_{h+1}(s_{h+1}^i, a') + \max_{a'} Q_{h+1}^k(s_{h+1}^i, a') \Big|$$

$$\leq 4H \Big| \widehat{Q}_h(s_h^i, a_h^i) - Q_h^k(s_h^i, a_h^i) - \widehat{r}_h(s_h^i, a_h^i) + r_h^k(s_h^i, a_h^i)$$

$$- \max_{a'} \widehat{Q}_{h+1}(s_{h+1}^i, a') + \max_{a'} Q_{h+1}^k(s_{h+1}^i, a') \Big|$$

$$+ 4H \Big| (\mathcal{T}_h^{\widehat{r}} \widehat{Q}_{h+1})(s_h^i, a_h^i) - (\mathcal{T}_h^{r^k} Q_{h+1}^k)(s_h^i, a_h^i) - \widehat{r}_h(s_h^i, a_h^i) + r_h^k(s_h^i, a_h^i)$$

$$- \max_{a'} \widehat{Q}_{h+1}(s_{h+1}^i, a') + \max_{a'} Q_{h+1}^k(s_{h+1}^i, a') \Big|$$

$$\leq 24 H \rho.$$

With the above bounds, we can obtain that

$$\sum_{i=0}^{k-1} \mathbb{E} \left[ \left( Q_h^k(s_h^i, a_h^i) - (\mathcal{T}_h^{r^k} Q_{h+1}^k)(s_h^i, a_h^i) \right)^2 \Big| \pi^i \right]$$

$$\leq 2 \sum_{i=0}^{k-1} X_h^i(Q^k, r^k) + 82 H^3 \log \left( 2KH \max_{h \in [H]} \mathcal{N}_\rho(\mathcal{Q}_h) \mathcal{N}_\rho(\mathcal{R}_h)/\delta \right) + 54 k H \rho.$$

According to the definition of $\mathrm{BE}^k$, we have that

$$\mathrm{BE}^k(Q^k)$$

$$= \sum_{h=1}^{H} \sum_{i=0}^{k-1} \left( Q_h^k(s_h^i, a_h^i) - r_h^k(s_h^i, a_h^i) - \max_{a'} Q_{h+1}^k(s_{h+1}^i, a') \right)^2$$

$$- \inf_{Q' \in \mathcal{Q}} \sum_{h=1}^{H} \sum_{i=0}^{k-1} \left( Q_h'(s_h^i, a_h^i) - r_h^k(s_h^i, a_h^i) - \max_{a'} Q_{h+1}^k(s_{h+1}^i, a') \right)^2$$

$$\overset{(a)}{\geq} \sum_{h=1}^{H} \sum_{i=0}^{k-1} X_h^i(Q^k, r^k)$$

$$\geq \frac{1}{2} \sum_{h=1}^{H} \sum_{i=0}^{k-1} \mathbb{E} \left[ \left( Q_h^k(s_h^i, a_h^i) - (\mathcal{T}_h^{r^k} Q_{h+1}^k)(s_h^i, a_h^i) \right)^2 \Big| \pi^i \right]$$

$$- 41 H^4 \log \left( 2KH \max_{h \in [H]} \mathcal{N}_\rho(\mathcal{Q}_h) \mathcal{N}_\rho(\mathcal{R}_h)/\delta \right) - 27 k H^2 \rho.$$

Inequality (a) follows Assumption 3 that $\mathcal{T}_h^{r^k} Q_{h+1}^{r^k} \in \mathcal{Q}_h$ We complete the proof.

## B.7  Technical Lemmas

**Lemma 6** (Freedman's inequality [3]). *Let $(X_t)_{t \leq T}$ be a real-valued martingale difference sequence adapted to filtration $\mathcal{F}_t$, and let $\mathbb{E}_t[\cdot] = \mathbb{E}[\cdot \mid \mathcal{F}_t]$. If $|X_t| \leq R$ almost surely, then for any $\eta \in [0, \frac{1}{R}]$ it holds that with probability at least $1 - \delta$,*

$$\sum_{t=1}^{T} X_t \leq \eta \sum_{t=1}^{T} \mathbb{E}_{t-1}[X_t^2] + \frac{\log(1/\delta)}{\eta}.$$

**Lemma 7.** *For any reward functions $r, \widehat{r}$, we have that*

$$\forall (s, a, h) \in \mathcal{S} \times \mathcal{A} \times [H],\ \left| Q_h^{\star, r}(s, a) - Q_h^{\star, \widehat{r}}(s, a) \right| \leq \sum_{h'=h}^{H} \max_{s \in \mathcal{S}, a \in \mathcal{A}} |r_h(s, a) - \widehat{r}_h(s, a)|.$$

*Here $Q^{\star, r}$ is the optimal Q-value function of $r$.*

*Proof.* According to the Bellman optimality equation, we have that

$$\left| Q_h^{\star, r}(s, a) - Q_h^{\star, \widehat{r}}(s, a) \right|$$

$$= \left| r_h(s, a) - \widehat{r}_h(s, a) + \mathbb{E}_{s' \sim P_h(\cdot | s, a)} \left[ \max_{a' \in \mathcal{A}} Q_{h+1}^{\star, r}(s', a') - \max_{a' \in \mathcal{A}} Q_{h+1}^{\star, \widehat{r}}(s', a') \right] \right|$$

$$\leq |r_h(s, a) - \widehat{r}_h(s, a)| + \mathbb{E}_{s' \sim P_h(\cdot | s, a)} \left[ \left| \max_{a' \in \mathcal{A}} Q_{h+1}^{\star, r}(s', a') - \max_{a' \in \mathcal{A}} Q_{h+1}^{\star, \widehat{r}}(s', a') \right| \right].$$

We analyze the term $| \max_{a' \in \mathcal{A}} Q_{h+1}^{\star, r}(s', a') - \max_{a' \in \mathcal{A}} Q_{h+1}^{\star, \widehat{r}}(s', a')|$.

$$\max_{a' \in \mathcal{A}} Q_{h+1}^{\star, r}(s', a') - \max_{a' \in \mathcal{A}} Q_{h+1}^{\star, \widehat{r}}(s', a')$$

$$= Q_{h+1}^{\star, r}(s', a^1) - Q_{h+1}^{\star, \widehat{r}}(s', a^2)$$

$$\leq Q_{h+1}^{\star, r}(s', a^1) - Q_{h+1}^{\star, \widehat{r}}(s', a^1),$$

$$\max_{a' \in \mathcal{A}} Q_{h+1}^{\star, r}(s', a') - \max_{a' \in \mathcal{A}} Q_{h+1}^{\star, \widehat{r}}(s', a')$$

$$= Q_{h+1}^{\star, r}(s', a^1) - Q_{h+1}^{\star, \widehat{r}}(s', a^2)$$

$$\geq Q_{h+1}^{\star, r}(s', a^2) - Q_{h+1}^{\star, \widehat{r}}(s', a^2).$$

Here $a^1 \in \operatorname{argmax}_{a' \in \mathcal{A}} Q_{h+1}^{\star, r}(s', a'), a^2 \in \operatorname{argmax}_{a' \in \mathcal{A}} Q_{h+1}^{\star, \widehat{r}}(s', a')$. Thus, we can get that

$$\left| \max_{a' \in \mathcal{A}} Q_{h+1}^{\star, r}(s', a') - \max_{a' \in \mathcal{A}} Q_{h+1}^{\star, \widehat{r}}(s', a') \right| \leq \max_{a' \in \mathcal{A}} Q_{h+1}^{\star, r}(s', a') - Q_{h+1}^{\star, \widehat{r}}(s', a')$$

$$\leq \max_{a' \in \mathcal{A}} \left| Q_{h+1}^{\star, r}(s', a') - Q_{h+1}^{\star, \widehat{r}}(s', a') \right|. \qquad (8)$$

Then we have that $\forall (s, a) \in \mathcal{S} \times \mathcal{A}$,

$$\left| Q_h^{\star, r}(s, a) - Q_h^{\star, \widehat{r}}(s, a) \right|$$

$$\leq |r_h(s, a) - \widehat{r}_h(s, a)| + \mathbb{E}_{s' \sim P_h(\cdot | s, a)} \left[ \left| \max_{a' \in \mathcal{A}} Q_{h+1}^{\star, r}(s', a') - \max_{a' \in \mathcal{A}} Q_{h+1}^{\star, \widehat{r}}(s', a') \right| \right]$$

$$\leq |r_h(s, a) - \widehat{r}_h(s, a)| + \max_{s' \in \mathcal{S}, a' \in \mathcal{A}} \left| Q_{h+1}^{\star, r}(s', a') - Q_{h+1}^{\star, \widehat{r}}(s', a') \right|.$$

Applying the above recursion inequality repeatedly from $h' = h$ to $h' = H$ with $Q_{H+1}^{\star, r}(s, a) = Q_{H+1}^{\star, \widehat{r}}(s, a) = 0$ completes the proof. $\qquad \square$

**Lemma 8.** *For $a \geq 1$ and $\varepsilon \leq 1$, when $K \geq 4 \log(4a/\varepsilon)/\varepsilon^2$, we have that*

$$\sqrt{\frac{\log(aK)}{K}} \leq \varepsilon.$$

*Proof.* We consider the function $f(K) = \sqrt{\log(aK)/K}$ and calculate the gradient.

$$f'(K) = \frac{1}{2}\left(\frac{\log(aK)}{K}\right)^{-1/2}\left(\frac{1-\log(aK)}{K^2}\right).$$

When $K \geq 4\log(4a/\varepsilon)/\varepsilon^2 \geq 4$, we have that $f'(K) \leq 0$, implying that $f(K)$ is a monotonically decreasing function in this range. Then we have that

$$\sqrt{\frac{\log(aK)}{K}} \leq \sqrt{\frac{\log\left(4a\log(4a/\varepsilon)/\varepsilon^2\right)}{4\log(4a/\varepsilon)}}\varepsilon$$

$$= \sqrt{\frac{\log(4a/\varepsilon) + \log(\log(4a/\varepsilon)) + \log(1/\varepsilon)}{4\log(4a/\varepsilon)}}\varepsilon$$

$$\overset{(a)}{\leq} \sqrt{\frac{\log(4a/\varepsilon) + \log(4a/\varepsilon) + \log(1/\varepsilon)}{4\log(4a/\varepsilon)}}\varepsilon$$

$$\overset{(b)}{\leq} \varepsilon.$$

Inequality $(a)$ follows that $\log(x) \leq x+1$ and inequality $(b)$ follows that $a \geq 1$.

$\square$

## C  Implementation Details

### C.1  Implementation Details of OPT-AIL

**Reward Update.** As mentioned in Section 4.2, we choose $\psi(r)$ in Eq. (2) as the gradient penalty (GP) regularization of the reward model [6], which can help stabilize the online optimization process by enforcing 1-Lipschitz continuity of the reward model $r$. Here $\mathcal{D}^I$ is linear interpolations between the replay buffer $\mathcal{D}^k$ and expert demonstrations $\mathcal{D}^E$.

$$\psi(r) = \mathbb{E}_{\tau \sim \mathcal{D}^I}\left[\sum_{h=1}^{H}(\|\nabla r_h(s_h, a_h)\| - 1)^2\right]$$

**Policy Update.** Here we present the implementation details of policy updates. Firstly, to stabilize the training process, we refine the optimism regularization term by subtracting a baseline Q-value function from random policy $\mu \equiv \text{Unif}(\mathcal{A})$, which has been utilized in [29, 32]. Furthermore, recognizing that initial state samples can be limited and lack diversity, we employ both the replay buffer $\mathcal{D}^k$ and expert demonstrations $\mathcal{D}^E$ to compute the Q-value loss, which is a common data augmentation approach and has been validated in many deep AIL methods [28, 15, 56]. Incorporating these two enhancements, we reformulate the Q-value model training objective as follows.

$$\min_{Q \in \mathcal{Q}} \mathbb{E}_{\tau \sim \mathcal{D}^k \cup \mathcal{D}^E}\left[\sum_{h=1}^{H}\left(Q_h(s_h, a_h) - r_h^k - \overline{Q}_{h+1}(s_{h+1}, \pi^k)\right)^2\right]$$

$$-\lambda\mathbb{E}_{\tau \sim \mathcal{D}^k \cup \mathcal{D}^E}\left[\sum_{h=1}^{H}\left(Q_h(s_h, \pi^k) - Q_h(s_h, \mu)\right)\right].$$

### C.2  Architecture and Training Details

The experiments are conducted on a machine with 64 CPU cores and 4 RTX4090 GPU cores. Each experiment is replicated five times using different random seeds. For each task, we adopt online DrQ-v2 [66] to train an agent with sufficient environment interactions [1] and regard the resultant policy as the expert policy. Then we roll out this expert policy to collect expert demonstrations. The architecture and training details of OPT-AIL and all baselines are listed below.

---

[1]3M training steps for `Cheetah Run`, `Hopper Hop`, and `Walker Run`, and 1M training steps for other tasks.

**OPT-AIL:** Our codebase of OPT-AIL extends the open-sourced framework of IQLearn. We retain the structure and parameter design of the actor and critic from the original framework while employing SAC [17] with a fixed temperature for policy update. We also implement a discriminator with a similar architecture to the critic network, and additionally incorporate layer normalization and tanh activation before the output to improve training stability. A comprehensive enumeration of the hyperparameters of OPT-AIL is provided in Table 2.

**BC:** We implement BC based on our codebase. The actor model is trained using Mean Squared Error (MSE) loss over 10k training steps.

**PPIL:** We use the author's codebase, which is available at https://github.com/lviano/p2il.

**IQLearn:** We use the author's codebase, which is available at https://github.com/Div99/IQ-Learn.

**DAC:** We reproduce the DAC based on our codebase. Due to the difference in updating the discriminator compared to OPT-AIL, we refer to the official DAC implementation when reproducing the discriminator. We remove the layer normalization and the tanh activation function before the output, and find that this resulted in better performance.

**FILTER:** We use the author's codebase, which is available at https://github.com/gswamy98/fast_irl.

**HyPE:** We use the author's codebase, which is available at https://github.com/gswamy98/hyper.

We emphasize that for a fair comparison, all algorithms are implemented using the same codebase [2], with all hyperparameters kept consistent except for the gradient penalty coefficient. Specifically, in OPT-AIL, the gradient penalty coefficient is set to 1 for `Cartpole Swingup`, `Walker Walk`, and `Walker Stand`, and 10 for other tasks. For baselines, the gradient penalty coefficient is always set to 10 as provided by the authors. We also attempt to adjust this parameter for the baselines but find that the default parameters provided by the authors work well.

Table 2: OPT-AIL Hyper-parameters.

| Parameter | Value |
|---|---|
| discount ($\gamma$) | 0.99 |
| gradient penalty coefficient ($\beta$) | 1, 10 |
| optimism regularization coefficient ($\lambda$) | $10^{-3}$ |
| temperature ($\alpha$) | $10^{-2}$ |
| replay buffer size | $5 \cdot 10^5$ |
| batch size | 256 |
| optimizer | Adam |
| *Discriminator* | |
| learning rate | $3 \cdot 10^{-5}$ |
| number of hidden layers | 2 |
| number of hidden units per layer | 256 |
| activation | ReLU |
| *Actor* | |
| learning rate | $3 \cdot 10^{-5}$ |
| number of hidden layers | 2 |
| number of hidden units per layer | 256 |
| activation | ReLU |
| *Critic* | |
| learning rate | $3 \cdot 10^{-4}$ |
| number of hidden layers | 2 |
| number of hidden units per layer | 256 |
| activation | ReLU |

---

[2]The codebase of PPIL is consistent with IQLearn.

# D  Additional Experimental Results

In this section, we list the learning curves for 8 DMControl tasks with 4, 7, and 10 expert trajectories respectively. The corresponding results are depicted in Figure 3, Figure 4, and Figure 5. Here the x-axis is the number of environment interactions and the y-axis is the return. The solid lines are the mean of results while the shaded region corresponds to the standard deviation over 5 random seeds. Our results demonstrate that OPT-AIL consistently achieves better interaction sample efficiency than state-of-the-art (SOTA) deep AIL methods, across varying numbers of expert trajectories.

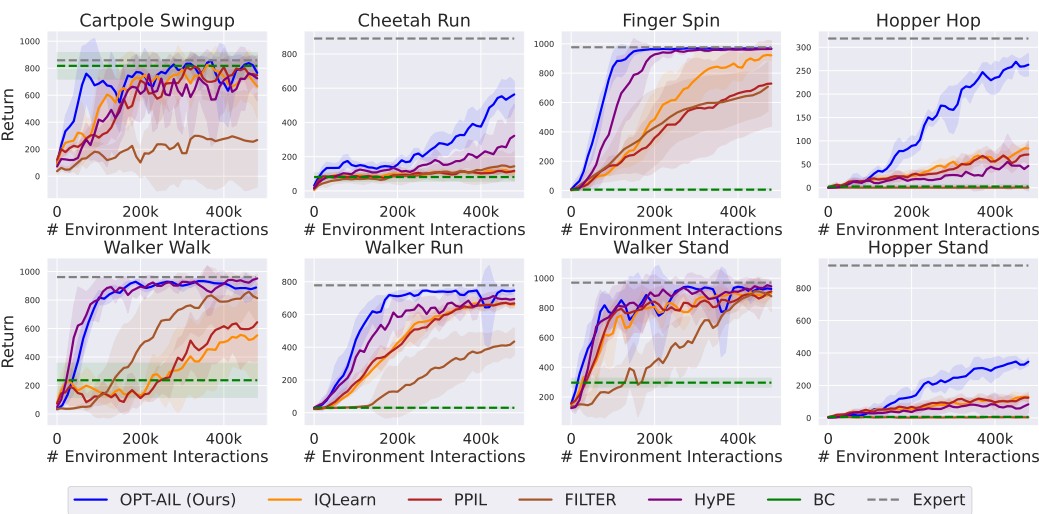

Figure 3: Learning curves on 8 DMControl tasks over 5 random seeds using 4 expert trajectories.

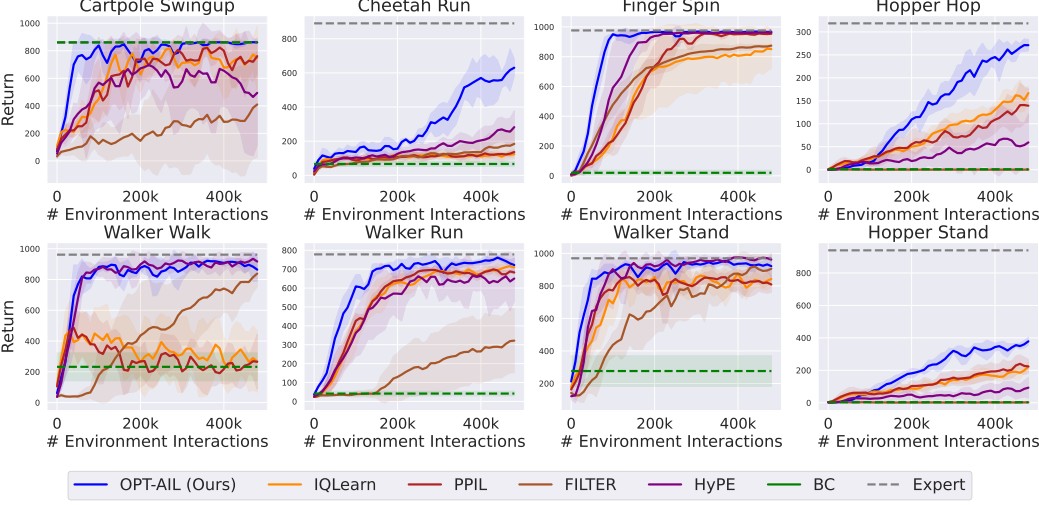

Figure 4: Learning curves on 8 DMControl tasks over 5 random seeds using 7 expert trajectories.

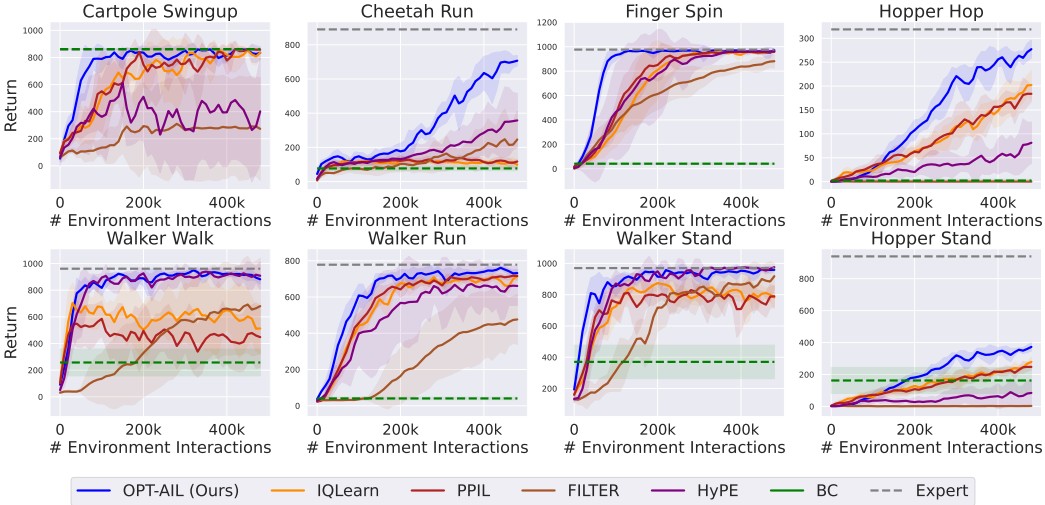

Figure 5: Learning curves on 8 DMControl tasks over 5 random seeds using 10 expert trajectories.

