# OpenReview forum: "Provably and Practically Efficient Adversarial Imitation Learning with General Function Approximation"
_NeurIPS.cc/2024/Conference — NeurIPS 2024 poster_

### Official Review · Reviewer_ntyv · 2024-06-18

**Soundness:** 3
**Presentation:** 3
**Contribution:** 2
**Rating:** 5
**Confidence:** 3

**Summary:**

This paper analyses the AIL problem in the context of general function approximation. Specifically, authors propose an algorithm which is both sample efficient and computationally efficient. Finally, the paper concludes with an empirical validation of the results.

**Strengths:**

- The paper analyses the AIL problem with general function approximation, which is interesting
- Author focus on both theoretical guarantees and practical implementation
- Authors validate the results empirically

**Weaknesses:**

- There are many typos. For instance:
- line 12 "near-expert"
- line 43 you say have extended, but [28] is older than [54]
- table 1 it should be "linear mixture"
- line 141 $\mathcal{R}_h$ does not represent a reward class defined in that way
- line 143,149 it should be "realizability"
- line 145 $\mathcal{Q}_h$ does not represent a class defined in that way
- line 190 I think that the value functions should have an hat
- ...

- The authors make many assumptions to solve the problem. In particular, assumptions 1,2,3, as well as assumption 4, which I am not sure how strong it is as structural assumption
- Algorithm 1 requires to keep in memory all the K policies collected so far, which are many according to Theorem 1. Thus, this stuff may be rather inefficient concerning the memory storage.

**Questions:**

- why your algorithm has an expert complexity which does not depend, differently from other algorithms, on the state-action space someway?
- At lines 246-248, when you say that your algorithm improves by an order of $\mathcal{O}(H^2)$ over BC, why do you think is so? Because of the structural assumptions that you added to the problem? Moreover, I do not understand why the rate of BC is $H^4$ instead of the common $H^2$.
- at lines 324-325, you say that a promising direction would be to try to achieve the optimal $H^{3/2}/\epsilon$ rate for the expert sample complexity. But who says that this is the optimal rate for the general function approximation setting? Authors [35] demonstrate that $H^{3/2}/\epsilon$ is optimal in the tabular setting thanks to accurate estimates of the transition model. But if you look at [34], authors say that the knowledge of the transition model does not allow to break the quadratic barrier in problems with continuous state space trivially, but have to devise something else. What do you think?

**Limitations:**

The authors have adequately addressed the limitations and potential negative societal impact of their work.

---

> ### Author Rebuttal · Authors · 2024-08-06
>
> Thank you for taking the time to review and check our paper, and for your insightful comments. The references mentioned in this response can be found in the global response section.
>
> **Question 1:** Typos.
>
> **Answer 1:** We have fixed these typos and thoroughly revised the paper.
>
> **Question 2:** The reviewer is not sure how strong assumptions 1-4 are as structural assumptions.
>
> **Answer 2:** We point out that assumptions 1-4 in this paper are **weaker** than those used in previous theoretical AIL works [R4-R7]. Here we verify that assumptions 1-4 in this paper are weaker than the linear MDP assumption in [R7], where both the transition model and the true reward are assumed to be linear.
>
> - **Assumption 1:** [R7] assumes the true reward is linear and applies a linear reward class, which implies reward realizability.
> - **Assumption 2:** [R7] employs a linear Q-value class. Additionally, in linear MDPs, the optimal Q-value function is linear [R11], which implies Q-value realizability.
> - **Assumption 3:** Claim 7.2 in [R10] indicates that linear MDP satisfies the Bellman Completeness assumption.
> - **Assumption 4:** Linear MDP is an instance of MDPs with low GEC [R8].
>
> Similarly, we can show that assumptions 1-4 are also weaker than the linear mixture MDP assumption in [R6]. Moreover, tabular MDP is an instance of linear MDPs and is therefore also stronger than assumptions 1-4. Overall, the assumptions used in this paper are weaker than those used in previous works [R4-R7].
>
> Furthermore, [R15] proves that the realizability and Bellman Completeness assumptions are necessary for RL with general function approximation (GFA). Since AIL requires solving a series of RL tasks, which is more complex than RL, we believe that the realizability and Bellman Completeness assumptions could be necessary for AIL with GFA. Nevertheless, it is an interesting direction to relax assumptions 1-4 in AIL with GFA.
>
> **Question 3:** Algorithm 1 requires to keep in memory all the K policies collected so far, which may be rather inefficient concerning the memory storage.
>
> **Answer 3:** Keeping historical policies is purely for sample complexity analysis and is widely applied in AIL theory works [R5, R7]. In experiments, we found that the last iterate policy already performs well, so there is no need to keep historical policies.
>
> **Question 4:** why your algorithm has an expert complexity which does not depend, differently from other algorithms, on the state-action space someway?
>
> **Answer 4:** The difference arises because our algorithm operates with **function approximation**, whereas other algorithms [R4, R5], whose expert complexities depend on the state-action space size $|\mathcal{S}| |\mathcal{A}|$, operates in the **tabular** setup. In AIL with **GFA**, our algorithm leverages a **reward class** $\mathcal{R}$ to infer the true reward, and this reward class can capture the **underlying structure** of the true reward. Thus, the expert complexity depends on the complexity of $\mathcal{R}$ (the covering number in this work) rather than on $|\mathcal{S}| |\mathcal{A}|$. For instance, if the true reward has a linear structure with dimension $d$ and $\mathcal{R}$ is selected as linear functions, we have $\log (\mathcal{N} (\mathcal{R})) = \mathcal{O}(d)$, meaning the expert complexity depends on $d$. In contrast, algorithms designed for the tabular setup recover the reward value for **each state-action pair independently**, causing their expert complexities to depend on the size of the state-action space.
>
> **Question 5:** Why your algorithm can improve by an order of $H^2$ over BC? Because of the structural assumptions? Why the rate of BC is $H^4$ instead of the common $H^2$?
>
> **Answer 5:** The improvement of OPT-AIL over BC arises from the fact that OPT-AIL can acquire additional transition information from **online interactions**, whereas BC operates solely in an **offline** manner. This improvement is **not** due to the structural assumptions. By interacting with the MDP, AIL can perform multi-step state-action distribution matching [R4, R5, R7], whereas BC is limited to single-step policy matching. This insight has been theoretically validated in both tabular [R4, R5] and linear [R7] settings. In this work, we verify this insight for GFA.
>
> Second, the $H^2$ rate for BC is achieved in the setup of **tabular MDPs** and **deterministic expert** [R16]. However, in the setting of **GFA** and **general expert** as studied in this work, Theorem 15.3 in [R10] shows that BC has an error bound $H^2 \sqrt{\frac{\log (|\Pi|)}{N}}$ when translating the result from an infinite horizon discounted MDP to a finite horizon MDP. This results in an $H^4$ expert complexity.
>
> **Question 6:** Who says that $H^{3/2} / \varepsilon$ is the optimal rate for the GFA setting? How to break the quadratic barrier for the GFA setting?
>
> **Answer 6:** First, we want to clarify that no existing work has established that the optimal rate for GFA is $\mathcal{O}(H^{3/2} / \varepsilon)$. We will clarify this point in the revised paper.
>
> Additionally, we agree with [R17] that breaking the quadratic barrier for GFA may necessitate additional assumptions about the imitation learning instance. Specifically, as discussed in [R5], to break the quadratic barrier, it might be necessary to roll out the BC policy to collect additional trajectories, enabling a more accurate estimation of the expert's state-action distribution. This approach depends on BC’s ability to achieve a low generalization error in its supervised learning (SL) task, allowing it to closely approximate the expert policy. To reach this objective, we may need to impose further assumptions on both the SL problem addressed by BC and the SL learner employed in BC.
>
> ---
> We hope that our responses can address your concerns satisfactorily. We would be grateful if you could re-evaluate our paper based on the above responses. We are also willing to address any further concerns, if possible.

---

> ### Comment · Reviewer_ntyv · 2024-08-09
>
> I thank the authors for the detailed and precise responses. I decide to keep my (positively biased) score, with (rather low) confidence.

---

> > ### Author Response · Authors · 2024-08-12
> >
> > We sincerely appreciate your constructive feedback throughout the review process. We will revise the paper according to your suggestions. We are pleased to know that you appreciate our responses, and we extend our gratitude for your positive score.

---

### Official Review · Reviewer_HVwx · 2024-06-30

**Soundness:** 3
**Presentation:** 3
**Contribution:** 3
**Rating:** 6
**Confidence:** 4

**Summary:**

This paper introduces optimization-based adversarial imitation learning (OPT-AIL), a novel method for online AIL with general function approximation. OPT-AIL combines online optimization for rewards and optimism-regularized Bellman error minimization for Q-value functions. Theoretically, it achieves polynomial expert sample complexity and interaction complexity, marking it as the first efficient AIL method with general function approximation. Practically, OPT-AIL simplifies implementation by requiring the optimization of only two objectives. Empirical results show OPT-AIL outperforms previous state-of-the-art deep AIL methods.

**Strengths:**

- This paper introduces OPT-AIL, a novel approach that addresses both theoretical and practical limitations of existing AIL methods by utilizing general function approximation.
- This paper provides both thoeretical and empirical results to validate the proposed algorithm.
- The error decomposition is new and provides a new point for understanding AIL.

**Weaknesses:**

- The complexity measure and main idea of algorithm is not entirely novel, which is based on GEC and a series of optimism-based work on general function approximation like GOLF.
- It seems that the paper mainly focus on the value-based hypothesis class and cannot incorporate the model-based ones directly.
- The discussions on errors $\varepsilon_{\rm opt}^r$ and $\varepsilon_{\rm opt}^Q$ is not sufficient (see Questions).

**Questions:**

- It is mentioned that one of the motivations of focusing on general function approximation is the implementation of neural networks in practice. How can neural network function class be included in the architecture? What’s the complexity of such classes?
- In the 3rd line of OPT-AIL, a no-regret algorithm is implemented to obtain the reward. Though author provides some explanations, it's still remain a bit confusing to me. Can authors elaborate more on this part, e.g., algorithm, optimization target, a brief recap of theoretical analysis (if these topics are well-dicussed in literature)?

**Limitations:**

I suggest the authors to include more dicussions on the primary techincal difficulties in contribution part while deriving their theoretical results for AIL.

---

> ### Author Rebuttal · Authors · 2024-08-06
>
> Thank you for taking the time to review our paper, and for your insightful comments. The references mentioned in this response can be found in the global response section.
>
> **Question 1:** The complexity measure and main idea of algorithm is not entirely novel, which is based on GEC and a series of optimism-based work on general function approximation like GOLF.
>
> **Answer 1:** We appreciate the series of works on GEC complexity and RL with general function approximation (GFA). However, our primary focus in this work is not the study of GEC complexity. Instead, our goal is to advance the understanding of adversarial imitation learning (AIL) with GFA. To achieve this, we establish a connection between AIL and RL with GFA, based on a new theoretical error decomposition. This connection enables the development of the first provably and practically efficient AIL approach for GFA.
>
> **Question 2:** It seems that the paper mainly focus on the value-based hypothesis class and cannot incorporate the model-based ones directly.
>
> **Answer 2:** Yes, this work focuses on value-based AIL. However, designing and analyzing a model-based AIL approach by leveraging the model optimism technique proposed in [R12, R13] is a valuable direction. We expect that the online optimization-based reward update used in this work would still be compatible with model-based approaches and the policy update would need to be redesigned.
>
> **Question 3:** The discussions on errors $\varepsilon^{r}\_{\text{opt}}$ and $\varepsilon^{Q}\_{\text{opt}}$ is not sufficient. Can authors elaborate more on the reward update, e.g., algorithm, optimization target, a brief recap of theoretical analysis?
>
> **Answer 3:** First, we elaborate on the reward update, which is based on **online** optimization.
>
> - **Algorithm & Optimization Target:** As discussed in lines 189-197, we design the reward update based on online optimization. The goal in online optimization is to minimize regret, defined as $\text{regret}:=\max_{x \in \mathcal{X}}\sum_{k=1}^K \mathcal{L}^{k} (x^k) - \mathcal{L}^{k} (x)$, with the relationship $\varepsilon^{r}\_{\text{opt}} = 1/K \cdot \text{regret}$. To achieve this, we can apply a no-regret algorithm, which can achieve sublinear regret (e.g., $\mathcal{O} (\sqrt{K})$), resulting in a small $\varepsilon_{\text{opt}}^r$ (e.g., $\mathcal{O} (1/\sqrt{K})$). For instance, if we use the no-regret algorithm Follow-the-Regularized-Leader (FTRL), it updates the reward by $r^{k} \leftarrow \arg\min_{r\in\mathcal{R}} \sum_{i=0}^{k-1} \mathcal{L}^{i} (r)+\beta\psi(r)$, where $\psi (r)$ is the regularization function.
>
> - **Theoretical Analysis:** The key step in the analysis is **error decomposition**, which decomposes the reward error into **optimization error** and **statistical error**. We can further upper bound the optimization error by $\varepsilon_{\text{opt}}^r$ and analyze the statistical error using concentration theory.
>
> Second, the Q-value update is based on **offline** optimization. In iteration $k$, we obtain $Q^{k}$ by solving the offline optimization problem $\min_{Q \in \mathcal{Q}} \mathcal{L}^k(Q)$, and the optimization error is defined as $\varepsilon^{Q}\_{\text{opt}} := \mathcal{L}^k (Q^k) - \min_{Q \in \mathcal{Q}} \mathcal{L}^k (Q)$.
>
> We will incorporate the above discussion in the revised paper.
>
> **Question 4:** How can neural network function class be included in the architecture? What’s the complexity of such classes?
>
> **Answer 4:** This work considers a general function class including neural networks (NNs). For NNs, the expert complexity and interaction complexity of OPT-AIL depend on the covering number of NNs. Theorem 3.3 in [R14] provides the covering number bound for NNs. Consider the L-layer NNs of
> $$
> \mathcal{F}:=\{f_{A}:f_{A}(s, a)=\sigma_L (A_L \cdots \sigma_1 (A_1 [s^T, a^T]^T))\}.
> $$
> Here, $A_1,\ldots,A_L$ are weight matrices with a spectral norm bound $b_s$ and a matrix norm bound $b_n$, while $\sigma_1, \ldots, \sigma_L$ are activation functions with Lipschitz coefficient $\rho$. The covering number is given by $\log (\mathcal{N} (\mathcal{F})) = \mathcal{O} ( (b_s\rho)^{2L} ( L (b_s/b_n)^{2/3}  )^{3} )$. Substituting this bound into the original expert complexity and interaction complexity yields the results for NNs.
>
> **Question 5:** Include more dicussions on the primary techincal difficulties in contribution part while deriving their theoretical results for AIL.
>
>
> **Answer 5:** The key theoretical challenge is that, unlike in RL, the reward functions in AIL are **stochastic** and exhibit **statistical dependence**, as they are learned from sampled expert demonstrations and environment interactions. This brings technical difficulties in analyzing both reward error and policy error.
>
> - **Analysis of Reward Error:** To analyze the reward error, we need to upper bound the statistical error that arises when using the empirical loss to approximate the expected one. However, due to the statistical dependence between reward functions, the standard concentration arguments for i.i.d. samples are not applicable. To address this issue, we construct a martingale difference sequence and apply a martingale concentration argument.
> - **Analysis of Policy Error:** To analyze the policy error, we need to analyze the difference between the empirical Bellman error, calculated from historical samples, and the true Bellman error for the recovered reward. The challenge arises because the recovered reward statistically depends on the historical samples, complicating the characterization of its concentration properties. To overcome this difficulty, we use a covering number argument and then design a martingale difference sequence to relate the empirical Bellman error to the true Bellman error.
>
> We will expand on the above discussion in the revised paper.
>
> ---
> We hope that the responses given above have effectively addressed your concerns. We are open to discussing any more questions you may have, if possible.

---

> > ### Comment · Reviewer_HVwx · 2024-08-08
> >
> > I thank the authors for the detailed response and I would suggest the authors to the authors to also incorporate answer in the revised version to better back up the claim in introduction. Besides, I think the difficulties and solution mentioned in answer 5, such as martingale concentration and covering number, is quite standard in RL rather than ``unlike”.  Overall, I am maintaining my original score and remain in favor of acceptance.

---

> > > ### Author Response · Authors · 2024-08-12
> > >
> > > Your valuable comments and feedback are deeply appreciated. We are pleased to know that our responses have addressed your questions, and we are committed to incorporating the above answers as we revise the paper. We extend our sincere gratitude for your positive evaluation.

---

### Official Review · Reviewer_bZiM · 2024-07-12

**Soundness:** 3
**Presentation:** 3
**Contribution:** 3
**Rating:** 7
**Confidence:** 4

**Summary:**

This paper studies adversarial imitation learning (AIL). From a theoretical perspective, it proposes a new algorithm OPT-AIL which works in the context of general function approximations, accompanied with a provable sample efficiency guarantee. The advantage of the proposed theoretical algorithm is that it can be easily adapted to a practical version based on neural network implementations.

**Strengths:**

**Orginality and Significance:**

1. The proposed algorithm is the first provably sample efficient online AIL under general function approximations, which is an important contribution to the theoretical understanding of imitation learning.
2. The proposed algorithm features an optimism-regularized Bellman-error minimization subproblem which makes the algorithm both provably sample efficient (for the online setup) and amenable to practical implementations based on neural networks.
3. Experimental results demonstrate the effectiveness of the proposed algorithm.

**Quality and Clarity:**

The presentation is quite clear. The theoretical results are sound and are well proved.

**Weaknesses:**

1. The idea and techniques in this paper seems direct given the existing theoretical works on AIL and RL with general function approximations especially [1] and [2].
2. The assumption of low generalized eluder coefficient [2] is from standard RL literature and is directly adapted here, without further explanations or discussions.

**References:**

[1] Zhihan Liu, Miao Lu, Wei Xiong, Han Zhong, Hao Hu, Shenao Zhang, Sirui Zheng, Zhuoran Yang, and Zhaoran Wang. Maximize to explore: One objective function fusing estimation, planning, and exploration. *Advances in Neural Information Processing Systems 36*, 2024.

[2] Han Zhong, Wei Xiong, Sirui Zheng, Liwei Wang, Zhaoran Wang, Zhuoran Yang, and Tong Zhang. A posterior sampling framework for interactive decision making. *arXiv*, 2211.01962, 2022.

**Questions:**

1. When the main theory translates to linear (or linear mixture) setups in AIL, how does the corresponding result compare with the previous arts?
2. Could the authors highlight the theoretical difficulties or novelties that arise from applying the idea of [1] to the setup of AIL?

**Limitations:**

Please see the weakness section and the question section above.

---

> ### Author Rebuttal · Authors · 2024-08-06
>
> We appreciate your time to review and provide positive feedback for our work. The references mentioned in this response can be found in the global response section.
>
> **Question 1:** The idea and techniques in this paper seems direct given the existing theoretical works on AIL and RL with general function approximations especially [1] and [2].
>
> **Answer 1:** We want to highlight that the theoretical analysis in this paper is **not** straightforward, given existing theoretical works on AIL and RL with general function approximation (GFA). The primary technical challenge arises from the fact that, in AIL, the reward functions are dynamic and stochastic, as they are learned from expert demonstrations and environment interactions in an online manner. This introduces statistical complexity in analyzing the joint learning processes of rewards and policies. Below, we provide a more detailed explanation of these theoretical challenges and our technical solutions.
>
> Unlike in RL, the reward functions in AIL are **stochastic** and exhibit **statistical dependence**, as they are learned from sampled expert demonstrations and environment interactions. This presents technical difficulties when analyzing both reward and policy errors.
> - **Analysis of Reward Error:** Analyzing reward error involves characterizing the statistical error that arises when using an empirical loss function to approximate the expected one. However, because the reward functions are statistically dependent, standard concentration arguments for i.i.d. samples are not applicable. This challenge is **unique** to AIL with **GFA** and is not encountered in existing theoretical works on AIL [R4-R6]. Previous studies focus on either tabular [R4, R5] or linear [R6] settings, which allow for error analysis in the state-action distribution space [R4, R5] or feature expectation space [R6] in a **reward-independent** manner. In contrast, our study on the GFA setting requires error analysis in the policy value space, which is inherently **reward-dependent**. To address the statistical dependence issue, we carefully construct a martingale difference sequence and apply a martingale concentration argument.
> - **Analysis of Policy Error:** Analyzing policy error requires analyzing the difference between the empirical Bellman error, calculated from historical samples, and the true Bellman error for the recovered reward. The challenge lies in the fact that the recovered reward statistically depends on these historical samples, complicating the characterization of concentration properties. To address this, we leverage a covering number argument and design a martingale difference sequence to relate the empirical Bellman error to the true Bellman error.
>
> We will elaborate on the theoretical challenges and our technical solutions in the revised paper.
>
> **Question 2:** The assumption of low generalized eluder coefficient [2] is from standard RL literature and is directly adapted here, without further explanations or discussions.
>
> **Answer 2:** In this work, the low generalized eluder coefficient (GEC) assumption is introduced to help control the policy error (as shown in Lemma 1) for AIL. We prove that the policy error can be upper bounded by the policy evaluation error (i.e., the term on the LHS of the low GEC assumption). The low GEC assumption ensures that the policy evaluation error can be controlled by the Bellman error on the dataset (i.e., the first term on the RHS of the low GEC assumption). By performing (regularized) Bellman error minimization, we can theoretically control the policy evaluation error, which in turn controls the policy error. We will elaborate on this low GEC assumption in the revised paper.
>
> **Question 3:** When the main theory translates to linear (or linear mixture) setups in AIL, how does the corresponding result compare with the previous arts?
>
> **Answer 3:** Here, we compare Theorem 1 with the previous result [R7] when applied to linear MDPs with dimension $d$. To adapt the main theory, we upper bound the terms $d_{\text{GEC}}$, $\mathcal{N}(\mathcal{R})$, and $\mathcal{N}(\mathcal{Q})$ in Theorem 1 by $d$.
>
> - For $d_{\text{GEC}}$, we can show that $d_{\text{GEC}} = \mathcal{O}(Hd)$. Specifically, $d_{\text{GEC}} = \mathcal{O}(H d_{\text{BE}})$ [R8], where $d_{\text{BE}}$ is the Bellman Eluder dimension. Furthermore, $d_{\text{BE}} = \mathcal{O}(d_{\text{BR}})$ [R9], where $d_{\text{BR}}$ is the Bellman rank, and $d_{\text{BR}} \leq d$ [R10]. Combining these bounds yields $d_{\text{GEC}} = \mathcal{O}(Hd)$.
> - Additionally, in linear MDPs, we use a linear reward class $\mathcal{R}$ and a linear Q-value class $\mathcal{Q}$. Based on the discussion following Corollary 16 in [R11], we have $\log(\mathcal{N}(\mathcal{R})) = \mathcal{O}(d)$ and $\log(\mathcal{N}(\mathcal{Q})) = \mathcal{O}(d)$.
>
> Using these bounds, we find that in linear MDPs, OPT-AIL achieves an expert sample complexity of $\widetilde{\mathcal{O}}(H^2d/\varepsilon^2)$ and an interaction complexity of $\widetilde{\mathcal{O}}(H^4d^2/\varepsilon^2)$. The expert sample complexity matches that of the BRIG approach [R7], while the interaction complexity improves upon BRIG [R7] by $\mathcal{O}(d)$. This improvement is due to the optimization-based optimism technique employed in our work, which offers better dimensional dependence than the bonus-based optimism used in [R7], as demonstrated in the RL literature [R9]. These findings confirm the sharpness of our main theory when applied to linear MDPs. We will include this discussion in the revised paper.
>
> **Question 4:** Could the authors highlight the theoretical difficulties or novelties that arise from applying the idea of [1] to the setup of AIL?
>
> **Answer 4:** We provide a detailed discussion on the theoretical challenges in **Answer 1**.
>
> ---
> We hope that the responses given above have effectively addressed your concerns. We are open to discussing any more questions you may have, if possible.

---

> > ### Comment · Reviewer_bZiM · 2024-08-09
> >
> > Thank you very much for your detailed answer to all my questions! I still appreciate the contributions of the work and I am in favor of the acceptance of the paper. I have no further questions and will remain my positive score.

---

> > > ### Author Response · Authors · 2024-08-12
> > >
> > > We sincerely appreciate your constructive feedback throughout the review process. We are fully committed to incorporating your suggestions as we revise the paper. We are glad to hear that you recognize the contributions of this work, and we are grateful for your positive score.

---

### Official Review · Reviewer_t2br · 2024-07-14

**Soundness:** 3
**Presentation:** 3
**Contribution:** 3
**Rating:** 7
**Confidence:** 4

**Summary:**

This paper explores the theory of adversarial imitation learning (AIL) using general function approximation. The authors introduce a novel approach called Optimization-Based AIL (OPT-AIL). OPT-AIL employs a no-regret subroutine for optimizing rewards and minimizes the optimism-regularized Bellman error for Q-value functions. The authors prove that OPT-AIL achieves polynomial expert sample complexity and interaction complexity, effectively imitating the expert. They also implement a practical version of OPT-AIL, demonstrating that it outperforms existing baseline methods across various environments.

**Strengths:**

1. Originality: This paper introduces the first provably efficient algorithm for adversarial imitation learning (AIL) with general function approximation.
2. Solid Mathematics: While I did not verify the proofs in the appendix, the algorithm appears standard, suggesting the proofs should be correct.
3. Good Writing: The paper is well-written, and the motivation is clear from the introduction and related work sections. Readers can easily understand the algorithm from sec 4.1 and the pseudo code.
4. Good Experimental Results: The practical version of OPT-AIL outperforms standard AIL baselines in various environments.

**Weaknesses:**

1. The practical algorithm itself is not highly innovative. The idea of running a no-regret algorithm to update the reward function is not new, and using an actor-critic framework for policy updates is also common.
2. The baselines compared are not SOTA algorithms for AIL/IRL. For instance, algorithms like FILTER (Swamy, Gokul, et al., "Inverse Reinforcement Learning Without Reinforcement Learning," ICML 2023) and HyPER (Ren, Juntao, et al., "Hybrid Inverse Reinforcement Learning," arXiv preprint, 2024) outperform IQ-Learn.

Minor issue:

3. For lines 270-272, the idea of using a no-regret algorithm for updating the reward function is not new. It has been explored and justified in previous work (such as Swamy, Gokul, et al., "Of Moments and Matching: A Game-Theoretic Framework for Closing the Imitation Gap," ICML 2021).

**Questions:**

1. Is it possible to add experiments comparing with FILTER and HyPER? I understand it is mainly a theory paper, but it would be good to add the latest SOTA algorithms to be the baselines.
2. Can FTRL directly be a justification for using off-policy update for reward function? Like, OGD is all a instance of FTRL but only update the reward function based on the current policy.

**Limitations:**

Yes, the authors have adequately addressed the limitations.

---

> ### Author Rebuttal · Authors · 2024-08-06
>
> Thank you for taking the time to review our paper and providing us with your valuable feedback. The references mentioned in this response can be found in the global response section.
>
> **Question 1:** The practical algorithm itself is not highly innovative. The idea of running a no-regret algorithm to update the reward function is not new, and using an actor-critic framework for policy updates is also common.
>
> **Answer 1:** We would like to emphasize that the novelty of our proposed practical algorithm, OPT-AIL, lies in its unique objective function for the Q-value, as presented in Eq. (3). Unlike previous practical AIL methods such as DAC and IQLearn, which update Q-value models by minimizing the **standard** Bellman error, OPT-AIL introduces **optimism-regularized** Bellman error minimization. This optimism-based regularization encourages exploration in unknown environments, a critical feature supported by our theoretical analysis. As a result, OPT-AIL benefits from theoretical guarantees with general function approximation. In contrast, previous practical AIL methods lack this optimistic mechanism, which may lead to the absence of such theoretical guarantees in those algorithms.
>
> **Question 2:** The baselines compared are not SOTA algorithms for AIL/IRL. For instance, algorithms like FILTER and HyPER outperform IQ-Learn.
>
> **Answer 2:** Thank you for bringing these two works to our attention. To address your concerns, we conducted additional experiments on FILTER and HyPE [R1]. For [R1], due to time constraints, we focused on evaluating the model-free algorithm HyPE. We chose this approach because HyPE demonstrates a similar return to the model-based method HyPER on locomotion tasks, as reported in their paper, but with significantly shorter training time (approximately 3 hours for HyPE compared to 30 hours for HyPER per run in our experiments). We tested FILTER and HyPE using the hyperparameters recommended in their respective papers.
>
> The detailed results are presented in Figures 1 and 2 of the submitted PDF. In terms of expert sample efficiency, OPT-AIL consistently matches or exceeds the performance of FILTER and HyPE. Regarding environment interaction efficiency, OPT-AIL achieves near-expert performance with fewer interactions compared to FILTER and HyPE. We believe this improvement could be due to the optimism-regularized Bellman error minimization technique employed in our approach, which facilitates more efficient exploration in the environment. In the revised paper, we will provide a comprehensive evaluation of FILTER, HyPE, and HyPER, and include these additional experimental results.
>
> **Question 3:** For lines 270-272, the idea of using a no-regret algorithm for updating the reward function is not new. It has been explored and justified in previous work (such as Swamy, Gokul, et al., "Of Moments and Matching: A Game-Theoretic Framework for Closing the Imitation Gap," ICML 2021).
>
> **Answer 3:** We want to clarify that we do **not** claim that the application of a no-regret algorithm for reward updates is novel. Instead, in lines 270-272, we establish a **connection** between the popular off-policy reward learning and the no-regret algorithm FTRL. To the best of our knowledge, this connection is new to the imitation learning literature and offers a partial explanation for the good practical performance of off-policy reward learning. We will make this point clearer in the revised paper and include a discussion of the related work [R2].
>
> **Question 4:** Can FTRL directly be a justification for using off-policy update for reward function? Like, OGD is all a instance of FTRL but only update the reward function based on the current policy.
>
> **Answer 4:** We establish a connection between off-policy reward learning and the Follow-The-Regularized-Leader (FTRL) algorithm by noting that **they share the same main optimization objective**. Specifically, the optimization objective for off-policy reward learning can be expressed as:
> $$
> \min_{r \in \mathcal{R}} \mathbb{E}\_{\tau \sim \mathcal{D}^k} \left[ \sum_{h=1}^H r_h (s^i_h, a^i_h) \right] - \mathbb{E}\_{\tau \sim \mathcal{D}^{\text{E}}} \left[ \sum_{h=1}^H r_h (s^i_h, a^i_h) \right]
> $$
> where $\mathcal{D}^k$ represents the replay buffer containing all historical samples. For FTRL in OPT-AIL, the optimization objective can be formulated as:
> $$
> \min_{r \in \mathcal{R}} \sum_{i=0}^{k-1} \mathcal{L}^i (r) + \beta \psi (r) \Leftrightarrow \min_{r \in \mathcal{R}} k \left( \mathbb{E}\_{\tau \sim \mathcal{D}^k} \left[ \sum_{h=1}^H r_h (s^i_h, a^i_h) \right] - \mathbb{E}\_{\tau \sim \mathcal{D}^{\text{E}}} \left[ \sum_{h=1}^H r_h (s^i_h, a^i_h) \right] \right) + \beta \psi (r).
> $$
> From these equations, it is evident that off-policy reward learning and FTRL share the same main objective. This insight suggests that viewing off-policy reward learning through the lens of FTRL can help explain its good practical performance. However, it’s important to note that while off-policy reward learning shares this objective with FTRL, in practice, it typically involves taking several gradient steps rather than fully optimizing this objective. As such, additional analysis is required to provide a complete theoretical explanation for off-policy reward learning, which is beyond the scope of this work.
>
> Finally, we would like to clarify that OGD is an instance of FTRL only when the loss functions $\\{ \mathcal{L}^i (r) \\}_{i=0}^K$ are linear [R3]. However, in practice, when reward models are parameterized by neural networks, the loss functions become non-linear, meaning that OGD is no longer an instance of FTRL in such cases.
>
> We greatly appreciate your question and welcome any further inquiries.
>
> ---
> We hope that our responses can address your concerns satisfactorily. We would be grateful if you could re-evaluate our paper based on the above responses. We are
> also willing to address any further concerns, if possible.

---

> ### Comment · Reviewer_t2br · 2024-08-10
>
> Thank you for your detailed response and for adding the experiments.
>
> Regarding Q2: I appreciate the additional experiments. The results are quite promising. For interaction efficiency, based on the plot, I would suggest describing the results as "competitive with HyPE, demonstrating more interaction-efficient learning in some environments." As for FILTER, you could try experiments in scenarios where exploration is particularly challenging in your final version, such as antmaze. Overall, I find the experimental results to be strong.
>
> Regarding Q3: I still believe that the point you're making is somewhat known in the field. It might be worth reviewing other papers that use FTRL for reward updates to avoid overstating the novelty of your contribution.
>
> Regarding Q4: Thank you for your response. I would recommend including this discussion in the final version of the paper.
>
> Overall, I think this is a strong theory paper with practical algorithms and solid experimental results. I’d like to raise my score to 7.

---

> > ### Author Response · Authors · 2024-08-12
> >
> > Thank you very much for your insightful comments and feedback. We will supplement these experimental results and revise the paper according to your suggestions. We are pleased to learn that our responses have addressed your concerns, and we deeply appreciate your reconsideration of the score.

---

### Author Rebuttal · Authors · 2024-08-06

Here we list all the references that appeared in the responses to reviewers.

References:

[R1] Juntao Ren et al. "Hybrid inverse reinforcement learning." arXiv: 2402.08848.

[R2] Gokul Swamy et al. "Of moments and matching: A game-theoretic framework for closing the imitation gap." ICML 2021.

[R3] Elad Hazan. "Introduction to online convex optimization." Foundations and Trends in Optimization, 2016.

[R4] Lior Shani et al. "Online apprenticeship learning." AAAI 2022.

[R5] Tian Xu et al., "Provably efficient adversarial imitation learning with unknown transitions.", UAI 2023.

[R6] Zhihan Liu et al. "Provably efficient generative adversarial imitation learning for online and offline setting with linear function approximation." ICML 2022.

[R7] Luca Viano et al. "Better imitation learning in discounted linear MDP." 2024.

[R8] Han Zhong et al. "A posterior sampling framework for interactive decision making." arXiv: 2211.01962.

[R9] Chi Jin et al. "Bellman eluder dimension: New rich classes of rl problems, and sample-efficient algorithms." NeurIPS 2021.

[R10] Alekh Agarwal et al., “Reinforcement learning: Theory and algorithms.”, 2019.

[R11] Chi Jin et al. "Provably efficient reinforcement learning with linear function approximation." COLT 2020.

[R12] Simon S. Du et al. "Bilinear classes: A structural framework for provable generalization in rl." ICML 2021.

[R13] Zhihan Liu et al. "Maximize to explore: One objective function fusing estimation, planning, and exploration." NeurIPS 2023.

[R14] Peter L. Bartlett et al. "Spectrally-normalized margin bounds for neural networks." NeurIPS 2017.

[R15] Dylan J. Foster et al. "Offline reinforcement learning: Fundamental barriers for value function approximation." COLT 2022.

[R16] Nived Rajaraman et al. "Toward the fundamental limits of imitation learning." NeurIPS 2020.

[R17] Nived Rajaraman et al. "On the value of interaction and function approximation in imitation learning." NeurIPS 2021.

---

### Decision · Program_Chairs · 2024-09-25

**Decision:**

Accept (poster)

**Comment:**

The paper proposes OPT-AIL, a novel adversarial imitation learning (AIL) algorithm with general function approximation, featuring both theoretical guarantees and empirical validation. Reviewers appreciated the solid theoretical contributions, good experimental results, and clear writing.

Reviewer t2br praised the theoretical novelty and experimental performance, suggesting minor improvements and raised the scores towards acceptance. Meanwhile, Reviewer bZiM found the paper strong and maintained a positive evaluation.

Reviewer HVwx  acknowledged the challenges addressed but noted that some techniques are standard in RL.

Reviewer ntvy pointed out some typos and assumptions but agreed with the solid contributions, but still staying slightly positive.

With reviewers in favor of the paper due to its technical soundness and practical results, I recommend acceptance of this paper as a poster presentation.